# ESTIMATION OF CONCEPT EXPLANATIONS SHOULD BE UNCERTAINTY AWARE

## ABSTRACT

Model explanations are very valuable for interpreting and debugging prediction models. We study a specific kind of global explanations called Concept Explanations, where the goal is to interpret a model using human-understandable concepts. Recent advances in multi-modal learning rekindled interest in concept explanations and led to several label-efficient proposals for estimation. However, existing estimation methods are unstable to the choice of concepts or dataset that is used for computing explanations. We observe that instability in explanations is due to high variance in point estimation of importance scores. We propose an uncertainty aware Bayesian estimation method, which readily improved reliability of the concept explanations. We demonstrate with theoretical analysis and empirical evaluation that explanations computed by our method are more reliable while also being label-efficient and faithful.

## 1 INTRODUCTION

With an ever increasing complexity of ML models, there is an increasing need to explain them. Concept-based explanations are a form of interpretable methods that explain predictions using high-level and semantically meaningful concepts (Kim et al., 2018). They are aligned with how humans communicate their decisions (Yeh et al., 2022) and are shown (Kim et al., 2018; 2023b) to be more preferable over explanations using salient input features (Ribeiro et al., 2016; Selvaraju et al., 2017) or salient training examples (Koh & Liang, 2017). Concept explanations also show potential in scientific discovery (Yeh et al., 2022) and for encoding task-specific prior knowledge (Yuksekgonul et al., 2022).

Concept explanations explain a pretrained prediction model by estimating the importance of concepts using two human-provided resources (1) a list of potentially relevant concepts for the task, (2) a dataset of examples usually referred to as the probe-dataset. Estimation usually proceeds in two steps (a) compute the log-likelihood of concept given an example called concept activations, and (b) aggregate their local activation scores into a globally relevant explanation. For example, the concept *wing* is considered important if the information about the concept is encoded in all examples of the *plane* class in the dataset. Owing to example-agnostic and classifier-level nature of concept explanations they are easy to interpret and have witnessed wide recognition in diverse applications (Yeh et al., 2022).

Despite their easy interpretation, concept explanations are known to be unreliable and data expensive. Ramaswamy et al. (2022a) showed that existing estimation methods are sensitive to the choice of concept set and dataset raising concerns over their interpretability. Another major limitation of concept-based explanation is the need for datasets with concept annotations in order to specify the concepts. Increasingly popular multi-modal models such as CLIP (Radford et al., 2021) present an exciting alternate direction to specify relevant concepts, especially for common image applications through their text description. Recent work has explored using multi-modal models for training concept-bottleneck models (Oikarinen et al., 2023; Yuksekgonul et al., 2022; Moayeri et al., 2023), but they are not yet evaluated for generating post-hoc concept explanations.

Our objective is to generate reliable concept explanations without requiring datasets with concept annotations. We begin by observing that existing estimation methods do not model noise in the estimation pipeline leading to high variance and unreliable explanations. We identify at least two causes of uncertainty (Section 4.1 presents more concrete scenarios) leading to unreliable explanations (1)

When a concept is missing from the probe-dataset, we cannot estimate its importance with confidence. Reporting uncertainty over estimated importance of a concept can thus help the user make a more informed interpretation, (2) When a concept is hard or irrelevant to the task their corresponding activations predicted from the representation layer of the model-to-be-explained are expected to be noisy. For example, it is harder to recognise the concept *whiskers* when compared with the concept *wings*. The noise or uncertainty in concept activations either due to their absence, hardness, or relevance if not modelled cascades into noise in explanations. Appreciating the need to model uncertainty, we present an estimator called Uncertainty-Aware Concept Explanations (U-ACE), which we show is instrumental in improving reliability of explanations.

**Contributions.**  • We motivate the need for modeling uncertainty for faithful estimation of concept explanations. • We propose a Bayesian estimation method called U-ACE that is both label-free and models uncertainty in the estimation of concept explanations. • We demonstrate the merits of our proposed method U-ACE through theoretical analysis and empirical evidence on two controlled datasets and two real-world datasets.

## 2 BACKGROUND AND MOTIVATION

We denote the model-to-be explained as $f : \mathbb{R}^D \to \mathbb{R}^L$ that maps D-dimensional inputs to L labels. Further, we use $f^{[l]}(\mathbf{x})$ to denote $l^{th}$ layer representation space and $f(\mathbf{x})[y]$ for $y \in [1, L]$ as the logit for the label $y$. Given a probe-dataset of examples $\mathcal{D} = \{\mathbf{x}^{(i)}\}_{i=1}^{N}$ and a list of concepts $\mathcal{C} = \{c_1, c_2, \dots, c_K\}$, our objective is to explain the pretrained model $f$ using the specified concepts. Traditionally, the concepts are demonstrated using potentially small and independent datasets with concept annotations $\{\mathcal{D}_c^k : k \in [1, K]\}$ where $\mathcal{D}_c^k$ is a dataset with positive and negative examples of the $k^{th}$ concept.

Concept-Based Explanations (CBE) estimate explanations in two steps. In the first step, they learn what are known as concept activation vectors that predict the concept from $l^{th}$ layer representation of an example. More formally, they learn the concept activation vector $v_k$ for $k^{th}$ concept by optimizing $v_k = \arg\min_v \mathbb{E}_{(x,y) \sim \mathcal{D}_c^{(k)}}[\ell(v^T f^{[l]}(\mathbf{x}), y)]$ where $\ell$ is the usual cross-entropy loss. The inner product of representation with the concept activation vector $v_k^T f^{[l]}(\mathbf{x})$ is usually referred to as concept activations. Various approaches exist to aggregate example-specific concept activations in to global example-agnostic explanations for the second step. Kim et al. (2018) computes sensitivity of logits to interventions on concept activations to compute what is known as CAV score per example per concept and report the fraction of examples in the probe-dataset with a positive CAV score as the global importance of the concept known as TCAV score. Zhou et al. (2018) proposed to decompose the classification layer weights as $\sum_k \alpha_k v_k$ and report the coefficients $\alpha_k$ as the importance score of the $k^{th}$ concept. We refer the reader to Yeh et al. (2022) for an in-depth survey.

**Data-efficient concept explanations.** A major limitation of traditional CBEs is their need for datasets with concept annotations $\{\mathcal{D}_c^1, \mathcal{D}_c^2, \dots\}$. In practical applications, we may wish to find important concepts among thousands of potentially relevant concepts, which is not possible without expensive data collection. Recent proposals (Yuksekgonul et al., 2022; Oikarinen et al., 2023; Moayeri et al., 2023) suggested using pretrained multi-modal models like CLIP to evade the data annotation cost for a related problem called Concept Bottleneck Models (CBM) (Koh et al., 2020). CBMs aim to train inherently interpretable model with a concept bottleneck. Although CBMs cannot generate explanations for a model-to-be-explained, a subset of methods propose to train what are known as Posthoc-CBMs using the representation layer of a pretrained task model for data efficiency. Given that Posthoc-CBMs base on the representation of a pretrained task model, we may use them to generate concept explanations. We describe briefly two such CBM proposals below.

Oikarinen et al. (2023) (O-CBM) estimates the concept activation vectors by learning to linearly project from the representation space of CLIP where the concept is encoded using its text description to the representation space of the model-to-be-explained $f$. It then learns a linear classification model on concept activations and returns the weight matrix as the concept explanation. Based on the proposal of Yuksekgonul et al. (2022), we can also generate explanations by training a linear model to match the predictions of model-to-be-explained directly using the concept activations of CLIP, which we denote by (Y-CBM).

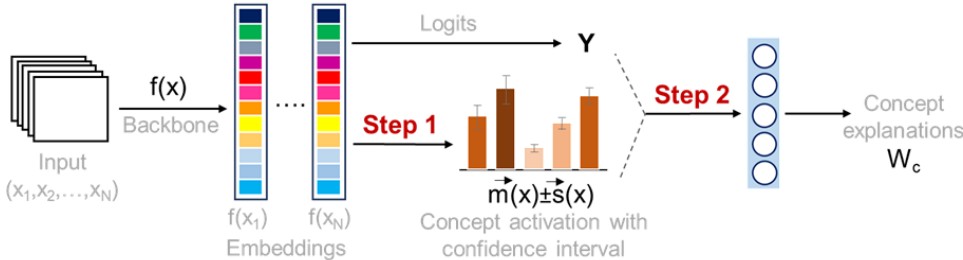

Figure 1: Our proposed estimator Uncertainty-Aware Concept Explanations

**Unreliable Explanations, a limitation.** Apart from data inefficiency, concept explanation methods are known to be unreliable. We observed critical reliability concerns with existing CBEs in the same spirit as the challenges raised in Ramaswamy et al. (2022a). As we demonstrate in Section 4.1, concept explanations for the same model-to-be-explained vary with the choice of the probe-dataset or the concept set bringing into question the reliability of explanations.

## 3 UNCERTAINTY-AWARE CONCEPT EXPLANATIONS

As summarized in the previous section, CBEs rely on concept activations for generating explanations. It is not hard to see that the activation score of a concept cannot be predicted confidently if the concept is hard/ambiguous or if it is not encoded by the model-to-be-explained. The noise in concept activations if not modeled cascades into the next step leading to poor explanations. Moreover, importance of a concept cannot be confidently estimated if it is missing from the probe-dataset, which must be informed to the user through confidence interval on the concept's estimated importance score. Motivated by the role of uncertainty for trustworthy explanations, we design our estimator.

Our approach has the following steps. (1) Estimate concept activations along with their error interval, (2) Aggregate concept activations and their confidence intervals in to a global concept explanation. We describe the estimation of concept activations and their error given an instance $\mathbf{x}$ denoted as $\vec{m}(\mathbf{x}), \vec{s}(\mathbf{x})$ respectively in Section 3.1. By definition, the true concept activation for a concept $k$ and instance $\mathbf{x}$ is in the range of $\vec{m}(\mathbf{x}) \pm \vec{s}(\mathbf{x})$ with a high probability. We describe the estimation of concept explanations in what follows using $\vec{m}(\mathbf{x}), \vec{s}(\mathbf{x})$, which is independent of how they are computed.

We compute explanations by fitting a linear regression model on the concept activations in the same spirit as many CBM methods because it is easier to incorporate the input noise in a regression model. Our objective is to learn linear model weights $W_c$ of size $L \times K$ (recall that L, K are the number of labels and concepts respectively) that map the concept activations to their logit scores, i.e. $f(\mathbf{x}) \approx W_c \vec{m}(\mathbf{x})$. Since the concept activations contain noise, we require that $W_c$ is such that predictions do not change under noise, that is $W_c[\vec{m}(\mathbf{x}) + \vec{s}(\mathbf{x})] \approx W_c \vec{m}(\mathbf{x}) \implies W_c \vec{s}(\mathbf{x}) \approx 0$. I.e. the inner product of each row ($\vec{w}$) of $W_c$ with $\vec{s}(\mathbf{x})$ must be negligible. For the sake of exposition, we analyse the solution of $\mathbf{y}^{th} \in [1, L]$ row $\vec{w}$ of $W_c$, which can be easily generalized to the other rows. We cast the bounded error constraint, i.e. $|\vec{w}^T \vec{s}(\mathbf{x})| \leq \delta$ for some small positive $\delta$ and for all the instances $\mathbf{x}$ in the probe-dataset, into a distributional prior over the weights. The prior over weights can then be easily accommodated in the Bayesian estimation of the posterior on weights.

$$|\vec{w}^T \vec{s}(\mathbf{x})| \leq \delta \quad \forall \mathbf{x} \in \mathcal{D} \implies |\vec{w}^T \epsilon| \leq \frac{\sum_{\mathbf{x} \in \mathcal{D}} |\vec{w}^T \vec{s}(\mathbf{x})|}{N} \leq \delta \text{ where } \epsilon \triangleq \frac{\sum_{x \in \mathcal{D}} \vec{s}(\mathbf{x})}{N}$$

$$|\vec{w}^T \epsilon| \leq \delta, \text{ for some small } \delta > 0 \text{ with high probability} \implies \vec{w}^T \epsilon \epsilon^T \vec{w} \approx \vec{w}^T \text{diag}(\epsilon \epsilon^T) \vec{w} \leq \delta^2$$

$$\implies -\frac{1}{2}(\vec{w} - \mathbf{0})^T S^{-1}(\vec{w} - \mathbf{0}) \text{ where } S^{-1} = \text{diag}(\epsilon \epsilon^T) \text{ is high when } \vec{w} \text{ satisfies the constraint}$$

$$\implies \mathcal{N}(\vec{w}; \mathbf{0}, \lambda S) \text{ is high for an appropriate } \lambda > 0 \implies \vec{w} \sim \mathcal{N}(\mathbf{0}, \lambda S)$$

We observe therefore that the weight vectors drawn from $\mathcal{N}(\mathbf{0}, \lambda \text{diag}(\epsilon \epsilon^T)^{-1})$ satisfy the invariance to input noise constraint with high probability. We now estimate the posterior on the weights after having observed the data with the prior on weights set to $\mathcal{N}(0, \lambda \text{diag}(\epsilon \epsilon^T)^{-1})$.

The posterior over weights has the following closed form(Salakhutdinov, 2011) where $C_X = [\vec{m}(\mathbf{x}_1), \vec{m}(\mathbf{x}_2), \ldots, \vec{m}(\mathbf{x}_N)]$ is a $K \times N$ matrix and $Y = [f(\mathbf{x}_1)[y], f(\mathbf{x}_2)[y], \ldots, f(\mathbf{x}_N)[y]]^T$ is an $N \times 1$ vector (derivation in Appendix A.1).

$$\Pr(\vec{w} \mid C_X, Y) = \mathcal{N}(\vec{w}; \mu, \Sigma) \qquad \text{where } \mu = \beta \Sigma C_X Y, \quad \Sigma^{-1} = \beta C_X C_X^T + \lambda^{-1} \text{diag}(\epsilon \epsilon^T) \quad (1)$$

$\beta$ is the inverse variance of noise in observations Y. We optimise both $\beta$ and $\lambda$ using MLE on $\mathcal{D}$ (more details in Appendix B). We could directly set the inverse of $\beta$ approximately 0 since there is no noise on the observations Y. Instead of setting $\beta$ to an arbitrary large value, we observed better explanations when we allowed the tuning algorithm to find a value of $\beta, \lambda$ to balance the evidence and noise.

**Sparsifying weights for interpretability.** Because a dense weight matrix can be hard to interpret, we induce sparsity in $W_c$ by setting all the values below a threshold to zero. The threshold is picked such that the accuracy on train split does not fall by more than $\kappa$, which is a positive hyperparameter.

The estimator shown in Equation 1 and details on how we estimate the noise in concept activations presented in the next section completes the description of our estimator. We call our estimator Uncertainty-Aware Concept Explanations (U-ACE) because it computes and models the uncertainty in concept activations. Algorithm 1 summarizes our proposed system.

## 3.1 ESTIMATION OF CONCEPT ACTIVATIONS AND THEIR NOISE

In this section, we discuss how we estimate $\vec{m}(\mathbf{x}), \vec{s}(\mathbf{x})$ using a pretrained multi-modal model. Recall that image-text multi-modal (MM) systems such as CLIP (Radford et al., 2021) can embed both images and text in a shared representation space, which enables one to estimate the similarity of an image to any phrase. This presents us an interesting solution approach of specifying a concept using its text description ($T_k$ for the $k^{th}$ concept) without needing concept datasets $\mathcal{D}_c^k$. We denote by $g(\bullet)$ the image embedding function of MM and $g_{text}(\bullet)$ the text embedding function.

Our objective is to estimate $\vec{m}(\mathbf{x}), \vec{s}(\mathbf{x})$ such that the true concept activation value is in the range $\vec{m}(\mathbf{x}) \pm \vec{s}(\mathbf{x})$. Two major sources of uncertainty in concept activations are due to (1) *epistemic uncertainty* arising from lack of information about the concept in the representation layer of the model-to-be-explained, (2) *data uncertainty* arising from ambiguity (because the concept is not clearly visible, see Appendix G.1 for some examples). We wish to estimate $\vec{s}(\mathbf{x})$ that is aware of both the forms of uncertainty.

We can obtain a point estimate for the activation vector of the $k^{th}$ concept $v_k$ such that $f(\mathbf{x})^T v_k \approx g(\mathbf{x})^T w_k$ (where $w_k = g_{text}(T_k)$) for all $\mathbf{x}$ in the probe-dataset $\mathcal{D}$ through simple optimization (Oikarinen et al., 2023; Moayeri et al., 2023). We may then simply repeat the estimation procedure multiple times to sample from the distribution of activation vectors and their corresponding concept activations. However, as shown empirically in Appendix G.1, $\vec{s}(\mathbf{x})$ estimated from random sampling is a poor measure of uncertainty.

We instead derive a closed form for $\vec{m}(\mathbf{x}), \vec{s}(\mathbf{x})$ based on the following intuition. The concept activations estimated using cos-sim($f(\mathbf{x}), v_k$) must intuitively be in the ballpark of $cos(\theta_k) =$ cos-sim($g(\mathbf{x}), w_k$) where cos-sim is the cosine similarity (Wikipedia, 2023a) (we switched from dot-products to cos-sim to avoid differences due to magnitude of the vectors). However, if the concept $k$ is not encoded in $f(\mathbf{x})$ or if it is ambiguous, the concept activations are expected to deviate by an angle $\alpha_k$, which is an error measure specific to the concept. Therefore, we expect the concept activations to be in the range of $cos(\theta_k \pm \alpha_k)$. The concept specific value $\alpha_k$ must account for uncertainty due to lack of knowledge (for eg. irrelevant concept) and due to ambiguity. In what follows, we present a specific measure for $\alpha_k$ and the closed form solution for $\vec{m}(\mathbf{x}), \vec{s}(\mathbf{x})$.

Borrowing from Oikarinen et al. (2023), we define $cos(\alpha_k)$ as
$\max_v[\text{cos-sim}(e(v, f, X), e(w_k, g, \mathcal{D}))]$ where $e(w_k, g, \mathcal{D}) \triangleq [w_k^T g(\mathbf{x}_1), \ldots, w_k^T g(\mathbf{x}_N)]^T$, and $e(v, f, \mathcal{D}) \triangleq [v^T f^{[-1]}(\mathbf{x}_1), \ldots, v^T f^{[-1]}(\mathbf{x}_N)]^T$.
We may just as well adopt any other measure for $\alpha_k$.

**Proposition 1.** *For a concept $k$ and a measure for $\alpha_k$, we have the following result when concept activations in $f$ for an instance $\mathbf{x}$ are computed as cos-sim($f(\mathbf{x}), v_k$) instead of $v_k^T f(\mathbf{x})$.*

$$\vec{m}(\mathbf{x})_k = cos(\theta_k)cos(\alpha_k), \quad \vec{s}(\mathbf{x})_k = sin(\theta_k)sin(\alpha_k)$$

*where $cos(\theta_k)$=cos-sim($g_{text}(T_k), g(\mathbf{x})$) and $\vec{m}(\mathbf{x})_k, \vec{s}(\mathbf{x})_k$ denote the $k^{th}$ element of the vector.*

The proof can be found in Appendix C. The mean and scale values above have a clean interpretation. If the model-to-be-explained ($f$) uses the $k^{th}$ concept for label prediction, the information about the concept is encoded in $f$ and we get a good fit, i.e. $cos(\alpha_k) \approx 1$, and a small error on concept activations. On the other hand, error bounds are large and concept activations are suppressed when the fit is poor, i.e. $cos(\alpha_k) \approx 0$. In Appendix G.1, we contrasted different methods for estimation of $\vec{s}(\mathbf{x})$. We observed from the empirical evaluation that U-ACE modeled both model and data uncertainty well.

## 3.2 THEORETICAL MOTIVATION

The motivation of this section is to demonstrate unreliability of concept explanations estimated using standard methods that do not model uncertainty during estimation. We particularly focus on unreliability due to misspecified concept set for the ease of analysis. In our study, we compared explanations generated using a standard linear estimator and U-ACE. Recall that posthoc-CBMs (O-CBM, Y-CBM), which are our primary focus for comparison, and they both estimate explanations by fitting a linear model on concept activations.

We present two scenarios with noisy concept activations. In the first scenario (over-complete concept set), we analyzed the estimation when the concept set contains many irrelevant concepts. We show that the likelihood of marking an irrelevant concept as more important than a relevant concept increases rapidly with the number of concepts when the explanations are estimated using a standard linear estimator that is unaware of the uncertainty. We also show that U-ACE do not suffer the same problem. In the second scenario (under-complete concept set), we analyzed the explanations when the concept set only includes irrelevant concepts, which should both be assigned a zero score ideally. We again show that standard linear model attributes a significantly non-zero score while U-ACE mitigates the issue. In Section 4.1, we confirm our theoretical findings with an empirical evaluation.

**Unreliable explanations due to over-complete concept set**. We analyze a simple setting where the output (y) is linearly predicted from the input (x) as $y = \mathbf{w}^T\mathbf{x}$. We wish to estimate the importance of some K concepts by fitting a linear estimator on concept activations. Where concept activations are computed as $\mathbf{w}_k^T\mathbf{x}$ using concept activation vectors ($\mathbf{w}_k$) that are distributed as $\mathbf{w}_k \sim \mathcal{N}(\mathbf{u}_k, \sigma_k^2 I), k \in [1, K]$.

**Proposition 2.** *The concept importance estimated by U-ACE when the input dimension is sufficiently large and for some $\lambda > 0$ is approximately given by $v_k = \frac{\mathbf{u}_k^T\mathbf{w}}{\mathbf{u}_i^T\mathbf{u}_k + \lambda\sigma_k^2}$. On the other hand, the importance scores estimated using Ordinary Least Squares (OLS) estimator under the same conditions is distributed as $v_k \sim \mathcal{N}(\frac{\mathbf{u}_k^T\mathbf{w}}{\mathbf{u}_k^T\mathbf{u}_k}, \sigma_k^2 \frac{\|w\|^2}{\|u_k\|^2})$.*

Proof of the result can be found in Appendix D. Based on the result, we can deduce the following result for a specific case of $u_k$s and $\sigma_k$s.

**Corollary 1.** *For the data setup of Proposition 2, the following results holds when $u_1 = \mathbf{w}, \sigma_1 \approx 0$ and $u_k^T\mathbf{w} = 0, \quad \forall k \in [2, K]$. Then the probability that the standard estimator returns the first concept as the most salient decreases exponentially with the number of concepts. On the other hand, the importance score assigned by U-ACE is 1 for the only relevant first concept and 0 otherwise.*

Derivation of the result can be found in Appendix A.2. We observe therefore that the probability of a random concept being estimated as more important than the relevant concept quickly converges to 1 with the number of random concepts K-1 when the distribution or uncertainty is not modeled. Sections 4.1, 5 demonstrate this phenomena in practice.

**Unreliable explanations due to under-complete concept set**. We now analyze explanations when the concept set only includes two irrelevant concepts. Consider normally distributed inputs $\mathbf{x} \sim \mathcal{N}(\mathbf{0}, I)$, and define two orthogonal unit vectors $\mathbf{u}, \mathbf{v}$. The concept activations: $c_1^{(i)}, c_2^{(i)}$ and label $y^{(i)}$ for the $i^{th}$ instance $\mathbf{x}^{(i)}$ are as defined below.

$$y^{(i)} = \mathbf{u}^T\mathbf{x}^{(i)}, \quad c_1^{(1)} = (\beta_1\mathbf{u} + (1 - \beta_1)\mathbf{v})^T\mathbf{x}^{(i)}, \quad c_2^{(i)} = (\beta_2\mathbf{u} + (1 - \beta_2)\mathbf{v})^T\mathbf{x}^{(i)}$$

If $\beta_1, \beta_2$ are very small, then both the concepts are expected to be unimportant for label prediction. However, we can see with simple working (Appendix E) that the importance scores computed by a

standard estimator are $\frac{1-\beta_2}{\beta_1-\beta_2}, \frac{1-\beta_1}{\beta_1-\beta_2}$, which are large because $\beta_1 \approx 0, \beta_2 \approx 0 \therefore \beta_1 - \beta_2 \approx 0$. We will now show that U-ACE estimates near-zero importance scores as expected.

**Proposition 3.** *The importance score, denoted $\eta_1, \eta_2$, estimated by U-ACE are bounded from above by $\frac{1}{N\lambda}$, where $\lambda > 0$ is a regularizing hyperparameter and N the number of examples.*

Proof can be found in Appendix E. It follows from the result that the importance scores computed by U-ACE are near-zero for sufficiently large value of $\lambda$ or N.

## 4 EXPERIMENTS

We evaluate U-ACE on two synthetic and two real-world datasets. We demonstrate how reliability of explanations is improved by U-ACE using a controlled study in Section 4.1. We make a quantitative assessment with known ground-truth on a controlled dataset in Section 5. Finally, we evaluate on two challenging real-world datasets with more than 700 concepts in Section 6.

**Baselines.** *Simple:* $W_c$ is estimated using lasso regression of ground-truth concept annotations to estimate logit values of $f$. *Simple* was also adopted in the past (Ramaswamy et al., 2022b;a) for estimating completeness of concepts. Other baselines are introduced in Section 2: *TCAV* (Kim et al., 2018), *O-CBM* (Oikarinen et al., 2023), *Y-CBM* based on (Yuksekgonul et al., 2022).

**Standardized comparison between importance scores.** The interpretation of the importance score varies between different estimation methods. For instance, the importance scores in TCAV is the fraction of examples that meet certain criteria while for other methods the importance scores are the weights from linear model that predicts logits. Further, *Simple* operates on binary concept annotations and *O-CBM, Y-CBM, U-ACE* on soft scores estimated using concept activation vectors. For this reason, we cannot directly compare importance scores or their normalized variants. We instead use negative scores to obtain a ranked list of concepts and assign to each concept an importance score given by its rank in the list normalized by number of concepts. Our sorting algorithm ranks any two concepts with same score by alphabetical order of their text description. In all our comparisons we use the rank score if not mentioned otherwise.

**Other experiment details.** For all our experiments, we used a Visual Transformer (with 32 patch size called "ViT-B/32") based pretrained CLIP model that is publicly available for download at https://github.com/openai/CLIP. We use $l = -1$, i.e. last layer just before computation of logits for all the explanation methods. U-ACE returns the mean and variance of the importance scores as shown in Algorithm 1, we use mean divided by standard deviation as the importance score estimated by U-ACE everywhere for comparison with other methods.

### 4.1 SIMULATED STUDY

In this section, we consider explaining a two-layer CNN model trained to classify between solid color images with pixel noise as shown in Figure 2. The colors red, green on the left are defined as label 0 and the colors blue, white on the right are defined as label 1. The model-to-be-explained is trained on a dataset with equal proportion of all colors, so we expect that all constituent colors of a label are equally important for the label. We specify a concept set with the four colors encoded by their literal name *red, green, blue, white*. U-ACE (along with others) 

Figure 2: Toy

attribute positive importance for *red, green* and negative or zero importance for *blue, white* when explaining label 0 using a concept set with only the four task-relevant concepts and when the probe-dataset is the same distribution as the the training dataset. However, quality of explanations quickly degrade when the probe-dataset is shifted or if the concept set is misspecified.

**Unreliability due to dataset shift.** We varied the probe-dataset to include varying population of different colors while keeping the concept set and model-to-be-explained fixed. We observed that importance of a concept estimated with standard CBEs varied with the choice of probe-dataset for the same underlying model-to-be-explained as shown in left and middle plots of Figure 3. Most methods attributed incorrect importance to the *red* concept when it is missing (left extreme of left

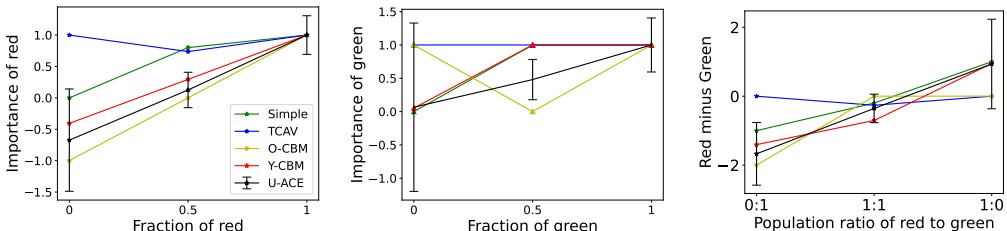

Figure 3: Left, middle plots show the importance of red and green concepts while the rightmost plot shows their importance score difference. U-ACE estimated large uncertainty in importance score when red or green concept is missing from the dataset as seen in the left of the left and middle plots. Also the difference in importance at either extreme in the right plot is not statistically significant.

plot), and similarly for the *green* concept (left extreme of middle plot). The explanations would have led the user to believe that *green* is more important than *red* or *red* is more important than *green* depending on the probe-dataset used as shown in the right most plot. Because U-ACE also informs the user of uncertainty in the estimated importance, we see that the difference in importance scores between the two colors at either extremes is not statistically significant as shown in the rightmost plot.

**Over-complete concept set**. We now evaluate the quality of explanations when the concept set is misspecified. More specifically, when the concept set is made over-complete by gradually expanding it to include common fruit names (Appendix F contains the full list), which are clearly irrelevant to the task. We obtain the explanations using an in-distribution probe-dataset that contains all colors in equal proportion. Figure 4 shows the score of most salient fruit concept with increasing number of fruit (nuisance) concepts on X-axis. We observe that U-ACE is far more robust to the presence of nuisance concepts. Robustness to irrelevant concepts is important because it allows the user to begin with a superfluous set of concepts and find their relevance to model-to-be-explained instead of requiring to guess relevant concepts, which is ironically the very purpose of using concept explanations. Appendix H presents and evaluates on an under-complete concept setting.

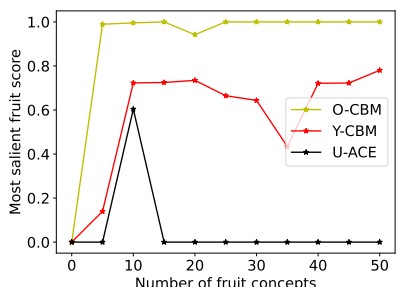

Figure 4: U-ACE is reliable even with overly complete concept set.

## 5 ASSESSMENT WITH KNOWN GROUND-TRUTH

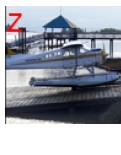

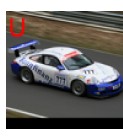

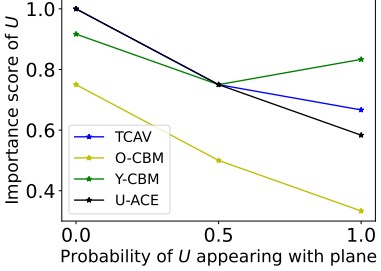

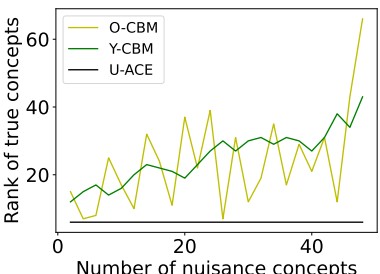

Figure 5: On the left is STL dataset with a spurious tag. In the middle is importance of a tag concept for three different model-to-be-explained. X-axis shows the probability of tag in the training dataset of model-to-be-explained. To the right is average rank of true concepts with irrelevant concepts (lower is better).

**Tree Farm**
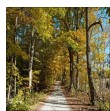
*Simple*: tree, field, bush
*O-CBM*: forest, pot, sweater
*Y-CBM*: field, forest, elevator
*U-ACE*: foliage, forest, grass

**Coast**
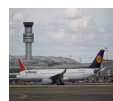
*Simple*: sea, water, river
*O-CBM*: sea, island, pitted
*Y-CBM*: sea, sand, towel rack
*U-ACE*: sea, lake, island

**Pasture**
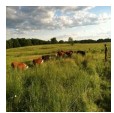
*Simple*: horse, sheep, grass
*O-CBM*: shaft, hoof, exhibitor
*Y-CBM*: field, grass, ear
*U-ACE*: grass, cow, banded

**Runway**
*Simple*: plane, field, sky
*O-CBM*: plane, fuselage, apron
*Y-CBM*: plane, clouds, candlestick
U-ACE: plane, windscreen, sky

Figure 6: Top-2 salient concepts plus any mistake (marked in red) from top-10 salient concepts for a scene-classification model estimated with PASCAL (left) or ADE20K (right) probe-dataset.

Our objective in this section is to establish that U-ACE generates faithful and reliable concept explanations. Subscribing to the common evaluation practice (Kim et al., 2018), we generate explanations for a model that is trained on a dataset with controlled correlation of a spurious pattern. We make a dataset using two labels from STL-10 dataset (Coates et al., 2011) *car, plane* and paste a tag *U* or *Z* in the top-left corner as shown in the left panel of Figure 5. The probability that the examples of *car* are added the *Z* tag is p and 1-p for the *U* tag. Similarly for the examples of *plane*, the probability of *U* is p and *Z* is 1-p. We generate three training datasets with p=0, p=0.5 and p=1, and train three classification models using 2-layer convolutional network. Therefore, the three models are expected to have a varying and known correlation with the tag, which we hope to recover from its concept explanation.

We generate concept explanations for the three model-to-be-explained using a concept set that includes seven car-related concepts and three plane-related concepts (Appendix F) along with the two tags *U, Z*. We obtain the importance score of the concept *U* with *car* class using a probe-dataset that is held-out from the corresponding training dataset (i.e. probe-dataset has the same input distribution as the training dataset). The results are shown in the middle plot of Figure 5. Since the co-occurrence probability of *U* with *car* class goes from 1, 0.5 to 0 for p=0, 0.5, 1, we expect the importance score of *U* should change from positive to negative as we move right. We note that U-ACE, along with others, show the expected decreasing importance of the tag concept. The result corroborates that U-ACE estimates a faithful explanation of model-to-be-explained while also being more reliable as elaborated below.

**Unreliability due to misspecified concept set.** In the same spirit as the previous section, we repeat the over-complete experiment of Section 4.1 and generated explanations as animal (irrelevent) concepts are added (Appendix F contains the full list). Right panel of Figure 5 shows the average rank of true concepts (lower the better). We note that U-ACE ranks true concepts highly even with 50 nuisance concepts.

## 6 REAL-WORLD EVALUATION

We expect that our reliable estimator to also generate higher quality concept explanations in practice. To verify the same, we generated explanations for a scene classification model with ResNet-18 architecture pretrained on Places365 (Zhou et al., 2017a), which is publicly available. Following the experimental setting of Ramaswamy et al. (2022a), we generate explanations when the probe-dataset is set to PASCAL (Chen et al., 2014) or ADE20K (Zhou et al., 2017b), which are both part of the Broden dataset (Bau et al., 2017b). The dataset contains images with dense annotations with more than 1000 attributes. We ignored around 300 attributes describing the scene since model-to-be-explained is itself a scene classifier. For the remaining 730 attributes, we defined a concept per attribute using literal name of the attribute. We picked 50 scene labels (Appendix F contains the full list) that have support of at least 20 examples in both ADE20K and PASCAL datasets.

We evaluate quality of explanations by their closeness to the explanations generated using the *Simple* baseline. *Simple* estimates explanation using true concept annotations and therefore its explanation must be the closest to the ground-truth. For the top-20 concepts identified by *Simple*, we compute the

average absolute difference in importance scores estimated using any estimation method and *Simple*. Table 1 presents the deviation in explanations averaged over all the 50 scene labels. Figure 6 shows the most salient concepts for four randomly picked scene labels. We observe from the figure that top-10 concepts identified by U-ACE seem more relevant to the scene when compared with Y-CBM and O-CBM. We also evaluated the explanation quality using a standard measure for comparing ranked lists, which is presented in Appendix F, which further confirms the dominance of U-ACE.

**Dataset shift.** Ramaswamy et al. (2022a) demonstrated with results the drastic shift in concept explanations for the same model-to-be-explained when using ADE20K or PASCAL as the probe-dataset. Explanations diverge partly because (a) population of concepts may vary between datasets thereby influencing their perceived importance when using standard methods, (b) variance in explanations. We have demonstrated that U-ACE estimated importance scores have low variance (shown in Section 3.2, 4.1) and attributes high uncertainty and thereby near-zero importance to concepts that are rare or missing from the probe-dataset (Section 4.1). For these reasons, we expect U-ACE to mitigate the data-shift problem. We confirm the same by estimating the average difference in importance scores estimated using ADE20K and PASCAL for different estimation techniques (where the average is only over salient concepts with non-zero importance). The results are shown in Table 2 and are inline with our prediction.

| Dataset↓ | TCAV | O-CBM | Y-CBM | U-ACE |
|---|---|---|---|---|
| ADE20K | 0.13 | 0.19 | 0.16 | **0.09** |
| PASCAL | 0.41 | 0.20 | 0.18 | **0.11** |

Table 1: *Evaluation of explanation quality.* Each cell shows the average absolute difference of importance scores for top-20 concepts estimated using *Simple*.

| Simple | TCAV | O-CBM | Y-CBM | U-ACE |
|---|---|---|---|---|
| 0.41 | 0.41 | 0.32 | 0.33 | **0.19** |

Table 2: *Effect of data shift.* Average absolute difference between concept importance scores estimated using ADE20K and PASCAL datasets for the same model-to-be-explained using different estimation methods.

## 7   RELATED WORK

**Concept Bottleneck Models** use a set of predefined human-interpretable concepts as an intermediate feature representation to make the predictions (Koh et al., 2020; Bau et al., 2017a; Kim et al., 2018; Zhou et al., 2018). CBM allows human test-time intervention which has been shown to improve overall accuracy (Barker et al., 2023). Traditionally, they require labelled data with concept annotations and typically the accuracy is worse than the standard models without concept bottleneck. To address the limitation of concept annotation, recent works have leveraged large pretrained multi-modal models like CLIP (Oikarinen et al., 2023; Yuksekgonul et al., 2022). There have also been efforts to enhance the reliability of CBMs by focusing on the information leakage problem (Havasi et al., 2022; Marconato et al., 2022), where the linear model weights estimated from concept activations utilize the unintended information, affecting the interpretability. Concept Embedding Models (CEM) (Espinosa Zarlenga et al., 2022) overcome the trade-off between accuracy and interpretability by learning high-dimensional concept embeddings. However, addressing the noise in the concept prediction remains underexplored. Collins et al. (2023) have studied human uncertainty in concept-based models and have shown the importance of considering uncertainty over concepts in improving the reliability of the model. Kim et al. (2023a) proposed the Probabilistic Concept Bottleneck Models (ProbCBM) and is closely related to our work. They too argue for the need to model uncertainty in concept prediction for reliable explanations. However, their method of noise estimation in concept activations requires retraining the model and cannot be applied directly when concept activations are estimated using CLIP. Moreover, they use simple MC sampling to account for noise in concept activations.

**Concept based explanations** use a separate probe dataset to first learn the concept and then explain through decomposition either the individual predictions or overall label features. Yeh et al. (2022) contains a brief summary of existing concept based explanation methods. Our proposed method is very similar to concept based explanations (CBE) (Kim et al., 2018; Bau et al., 2017a; Zhou et al., 2018; Ghorbani et al., 2019). Ramaswamy et al. (2022a) emphasized that the concepts learned are sensitive to the probe dataset used and therefore pose problems when transferring to applications that have distribution shift from the probe dataset. Moreover, they also highlight other drawbacks of existing CBE methods in that concepts can sometimes be harder to learn than the label itself (meaning the explanations may not be causal) and that the typical number of concepts used for ex-

planations far exceed what a typical human can parse easily. Achtibat et al. (2022) championed an explanation method that provides explanation highlighting important feature (answering "where") and what concepts are used for prediction thereby combining the strengths of global and local explanation methods. Choi et al. (2023) have built upon the current developments in CBE methods for providing explanations for out-of-distribution detectors. Wu et al. (2023) introduced the causal concept based explanation method (Causal Proxy Model), that provides explanations for NLP models using counterfactual texts. Moayeri et al. (2023) also used CLIP to interpret the representations of a different model trained on uni-modal data.

## 8 CONCLUSION

We studied concept explanation methods with a focus on data-efficient systems that exploit pre-trained multi-modal models. We demonstrated with simple examples the reliability challenge of existing estimators of concept explanations and motivated the need for modeling uncertainty in estimation and informing user the uncertainty in importance scores. Accordingly, we proposed an uncertainty-aware and data-efficient estimator called U-ACE, which readily yielded several benefits. We demonstrated the merits of our estimator through theoretical analysis, controlled study experiments and two challenging real-world evaluation with around 700 concepts. To the best of our knowledge, previous evaluations did not consider concept explanations with as many concepts. Our results showed that concept explanations estimated by U-ACE are more reliable.

**Limitations and Future Work** • The need and advantage when modeling uncertainty is also applicable when learning concept activations using datasets with concept annotations. However, our experimental setup is only focused on using CLIP for specifying concepts. • We did not model the uncertainty in CLIP's knowledge of a concept. Epistemic uncertainty due to CLIP when modelled may improve reliability further, which we leave for future work.

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

# Appendix

## A  MISCELLANEOUS

### A.1  DERIVATION OF POSTERIOR ON WEIGHTS

The result of posterior distribution of weights follows directly from the form of posterior under normal prior on weights as explained as Salakhutdinov (2011) (Slide 10). For the sake of completeness, we also derive the result below.

$$
\begin{aligned}
\Pr(\vec{w} \mid C_X, Y) &\propto \Pr(Y \mid C_X, \vec{w}) \Pr(\vec{w}) \\
&= \mathcal{N}(Y; C_X^T \vec{w}, \beta^{-1}) \mathcal{N}(\vec{w}; 0, S_0) \text{ where } S_0^{-1} = \lambda^{-1} \mathrm{diag}(\epsilon\epsilon^T) \\
&\propto \exp\left\{ -\frac{\beta}{2}(Y - C_X^T \vec{w})^T (Y - C_X^T \vec{w}) - \frac{1}{2}\vec{w}^T S_0^{-1} \vec{w} \right\} \\
&\propto \exp\left\{ -\frac{1}{2}\vec{w}^T [\beta C_X C_X^T + S_0^{-1}]\vec{w} - \beta(C_X Y)\vec{w} \right\}
\end{aligned}
$$

We see that the posterior also takes the form of normal distribution with $\Sigma^{-1} = \beta C_X C_X^T + S_0^{-1}$ and $\mu = \beta \Sigma C_X Y$.

### A.2  COROLLARY OF PROPOSITION 2

Following the result of Proposition 2, we have the following result on the $v_k$ estimated by U-ACE and the standard linear estimator.

**Corollary 2.** *For the data setup of Proposition 2, the following results holds when $u_1 = \mathbf{w}, \sigma_1 \approx 0$ and $u_k^T \mathbf{w} = 0, \quad \forall k \in [2, K]$. Then the probability that the standard estimator returns the first concept as the most salient decreases exponentially with the number of concepts. On the other hand, the importance score assigned by U-ACE is 1 for the only relevant first concept and 0 otherwise.*

*Proof.* Plugging in the values for the special case of $u_1 = w, \sigma_1 \approx 0$ and $u_k^T w = 0, k \geq 2$ in the closed form solution from Proposition 2, we have the following results for the standard linear estimator and U-ACE.

Solution of standard linear estimator: $v_1 = 1$ and $v_k \sim \mathcal{N}(0, \sigma_k^2 \frac{\|w\|}{\|u_k\|^2})$ for k$\geq 2$.
For the first concept to remain the most salient, rest of the K-1 concepts must have an importance score less than 1. Recall that the probability that a random variable z$\sim \mathcal{N}(\mu, \sigma^2)$ less than a value $z_0$ is $\Phi(\frac{z_0 - \mu}{\sigma})$ where $\Phi$ is the Cumulative Distribution Function of a standard normal distribution. Therefore the probability that all the K-1 concepts having a value less than 1 is $\prod_{k=2}^K \Phi(\frac{1-0}{\sigma_k \|w\|/\|u_k\|}) = \prod_{k=2}^K \Phi(\frac{\|u_k\|}{\sigma_k \|w\|})$. Since the probability is a product over K-1 quantities, it decreases exponentially with K.

Solution of U-ACE: $v_1 = 1, v_2, v_3, \cdots = 0$ follows directly from plugging in the values in to result of the proposition.

$\square$

### A.3  ALGORITHM

## B  MAXIMUM LIKELIHOOD ESTIMATION OF U-ACE PARAMETERS

The posterior on weights shown in Equation 1 has two parameters: $\lambda, \beta$ as shown below with $C_X$ and Y are array of concept activations and logit scores (see Algorithm 1).

$$
\vec{w} \sim \mathcal{N}(\mu, \Sigma) \qquad \text{where } \mu = \beta \Sigma C_X Y, \quad \Sigma^{-1} = \beta C_X C_X^T + \lambda^{-1} diag(\epsilon\epsilon^T)
$$

**Algorithm 1: Uncertainty-Aware Concept Explanations (U-ACE)**

---

**Require:** $\mathcal{D}=\{\mathbf{x}_1, \mathbf{x}_2, \ldots, \mathbf{x}_N\}$, $\mathcal{T} = \{T_1, T_2, \ldots, T_K\}$, f (model-to-be-explained), g (CLIP), $\kappa$

    **for** $y = 1, \ldots, L$ **do**
        Y = $[f(\mathbf{x})[y]$ for $\mathbf{x} \in \mathcal{D}^T]$                      ▷ Gather logits
        $C_X = [\vec{m}(\mathbf{x}_1), \ldots, \vec{m}(\mathbf{x}_N)]$, $\epsilon = \mathbb{E}_{\mathcal{D}}[\vec{s}(\mathbf{x})]$     ▷ Estimate $\vec{m}(\mathbf{x}), \vec{s}(\mathbf{x})$ (Section 3.1)
        $\vec{w}_y \sim \mathcal{N}(\mu_y, \Sigma_y)$ where $\mu_y, \Sigma_y$ from Equation 1          ▷ Estimate $\lambda, \beta$ using MLL
    **end for**
    $W_c = \text{sparsify}([\vec{\mu}_1, \vec{\mu}_2, \ldots \vec{\mu}_L], \kappa)$          ▷ Suppress less useful weights, Section 3
    **return** $W_c, [\text{diag}(\Sigma_1), \text{diag}(\Sigma_2), \ldots \text{diag}(\Sigma_L)]$

---

We obtain the best values of $\lambda$ and $\beta$ that maximize the log-likelihood objective shown below.

$$\lambda^*, \beta^* = \arg\max_{\lambda, \beta} \quad \mathbb{E}_Z[-\frac{\beta^2\|Y - (C_X + Z)^T\vec{w}(\lambda, \beta)\|^2}{2} + \log(\beta)]$$

where Z is uniformly distributed in the range given by error intervals
$$Z \sim Unif([-\vec{s}(\mathbf{x}_1), -\vec{s}(\mathbf{x}_2), \ldots,], [\vec{s}(\mathbf{x}_1), \vec{s}(\mathbf{x}_2), \ldots,])$$

We implement the objective using Pyro software library (Bingham et al., 2019) and Adam optimizer.

## C  PROOF OF PROPOSITION 1

We restate the result for clarity.
For a concept k and $cos(\alpha_k)$ defined as cos-sim$(e(v_k, f, \mathcal{D}), e(w_k, g, \mathcal{D}))$, we have the following result when concept activations in $f$ for an instance $\mathbf{x}$ are computed as cos-sim$(f(\mathbf{x}), v_k)$ instead of $v_k^T f(\mathbf{x})$.

$$\vec{m}(\mathbf{x})_k = cos(\theta_k)cos(\alpha_k), \quad \vec{s}(\mathbf{x})_k = sin(\theta_k)sin(\alpha_k)$$

where $\cos(\theta_k)$=cos-sim$(g_{text}(T_k), g(\mathbf{x}))$ and $\vec{m}(\mathbf{x})_k, \vec{s}(\mathbf{x})_k$ denote the $k^{th}$ element of the vector.

*Proof.* Corresponding to $v_k$ in $f$, there must be an equivalent vector $w$ in the embedding space of g.

$$cos(\alpha_k) = \text{cos-sim}(e(v_k, f, \mathcal{D}), e(w_k, g, \mathcal{D})) = \text{cos-sim}(e(w, g, \mathcal{D}), e(w_k, g, \mathcal{D}))$$

Denote the matrix of vectors embedded using $g$ by $G = [g(\mathbf{x}_1), g(\mathbf{x}_2), \ldots, G(\mathbf{x}_N)]^T$ a $N \times D$ matrix (D is the dimension of $g$ embeddings). Let U be a matrix with S basis vectors of size $S \times D$. We can express each vector as a combination of basis vectors and therefore $G = AU$ for a $N \times S$ matrix A.

Substituting the terms in the cos-sim expression, we have:

$$cos(\alpha_k) = \text{cos-sim}(Gw, Gw_k) = \text{cos-sim}(AUw, AUw_k)$$
$$= \frac{w^T U^T A^T A U w_k}{\sqrt{(w^T U^T A^T A U w)(w_k^T U^T A^T A U w_k)}}.$$

If the examples in $\mathcal{D}$ are diversely distributed without any systematic bias, $A^T A$ is proportional to the identity matrix, meaning the basis of G and W are effectively the same. We therefore have $cos(\alpha_k) = \text{cos-sim}(Gw, Gw_k) = \text{cos-sim}(Uw, Uw_k)$, i.e. the projection of $w, w_k$ on the subspace spanned by the embeddings have $cos(\alpha_k)$ cosine similarity. Since $w, w_k$ are two vectors that are $\alpha_k$ apart, an arbitrary new example $\mathbf{x}$ that is at an angle of $\theta$ from $w_k$ is at an angle of $\theta \pm \alpha_k$ from w. The cosine similarity follows as below.

$$cos(\theta) = \text{cos-sim}(w_k, g(\mathbf{x})) \implies \text{cos-sim}(w, g(\mathbf{x})) = cos(\theta \pm \alpha_k)$$
$$= cos(\theta)cos(\alpha_k) \pm sin(\theta)sin(\alpha_k)$$

Because $w$ is a vector in $g$ corresponding to $v_k$ in $f$, cos-sim$(w, g(\mathbf{x})) = $ cos-sim$(v_k, f(\mathbf{x}))$.    □

# D   PROOF OF PROPOSITION 2

The concept importance estimated by U-ACE when the input dimension is sufficiently large and for some $\lambda > 0$ is approximately given by $v_k = \frac{\mathbf{u}_k^T \mathbf{w}}{\mathbf{u}_i^T \mathbf{u}_k + \lambda \sigma_k^2}$. On the other hand, the importance scores estimated using vanilla linear estimator under the same conditions is distributed as $v_k \sim \mathcal{N}(\frac{\mathbf{u}_k^T \mathbf{w}}{\mathbf{u}_k^T \mathbf{u}_k}, \sigma_k^2 \frac{\|w\|^2}{\|u_k\|^2})$.

*Proof.* We use the known result that inner product of two random vectors is close to 0 when the number of dimensions is large, i.e. $u_i^T u_j \approx 0, i \neq j$.

**Result with vanilla estimator.** We first show the solution using vanilla estimator is distributed as given by the result above. We wish to estimate $v_1, v_2, \ldots$ such that we approximate the prediction of model-to-be-explained: $y = w^T \mathbf{x}$. We denote by $w_k$ sampled from the normal distributin of concept vectors. We require $w^T \mathbf{x} \approx \sum_k v_k w_k^T \mathbf{x}$. In effect, we are optimising for $v$s such that $\|w - \sum_k v_k w_k\|^2$ is minimized. We multiply the objective by $u_k$ and use the result that random vectors are almost orthogonal in high-dimensions to arrive at objective $\arg\min_{v_k} \|w_k^T w - v_k(w_k^T w_k)\|$. Which is minimized trivially when $v_k = \frac{w_k^T w}{\|w_k\|^2}$. Since $w_k$ is normally distributed with $\mathcal{N}(u_k, \sigma_k^2 I)$, $w_k^T w = (u_k + \epsilon)^T w$, $\epsilon \sim \mathcal{N}(0, I)$ is also normally distributed with $\mathcal{N}(u_k^T w, \sigma_k^2 \|w\|^2)$. We approximate the denominator with its average and ignoring its variance, i.e. $\|w_k\|^2 = \mathcal{N}(\|u_k\|^2, \sigma_k^2) \approx \|u_k\|^2$ which is when $\|u_k\|^2 >> \sigma^2$. We therefore have the result on distribution of $v_k$.

**Using U-ACE.** Similar to vanilla estimator, U-ACE optimizes $v_k$ using the following objective.

$$\ell = \arg\min_v \{\|w - \sum_k v_k u_k\|^2 + \lambda \sum_k \sigma_k^2 v_k^2\}$$

setting $\frac{\partial \ell}{\partial v_k} = 0$ and using almost zero inner product result above, we have

$$-u_k^T (w - \sum_j v_j u_j) + \lambda \sigma_k^2 v_k = 0$$

$$\implies v_k = \frac{u_k^T w}{\|u_k\|^2 + \lambda \sigma_k^2}$$

$\square$

# E   PROOF OF PROPOSITION 3

The importance score, denoted $v_1, v_2$, estimated by U-ACE are bounded from above by $\frac{1}{N\lambda}$, i.e. $v_1, v_2 = \mathcal{O}(1/N\lambda)$ where $\lambda > 0$ is a regularizing hyperparameter and N the number of examples.

*Proof.* We first show that the values of $v_1, v_2$ in closed form are as below before we derive the final result.

$$v_1 = \frac{\frac{S_1}{S_2}(1 - \beta_2)^2}{\frac{S_1}{S_2}(\beta_2^2(1-\beta_1)^2 + \beta_1^2(1-\beta_2)^2) + \lambda(1-\beta_1)(1-\beta_2)}$$

$$v_2 = \frac{\frac{S_1}{S_2}(1 - \beta_1)^2}{\frac{S_1}{S_2}(\beta_1^2(1-\beta_2)^2 + \beta_2^2(1-\beta_1)^2) + \lambda(1-\beta_1)(1-\beta_2)}$$

where $S_1 = \sum_i y_1$, $S_2 = \sum_i y_i^2$ and $\lambda > 0$ is a regularizing hyperparameter.

We then observe that if $\mathbf{x}$ is normally distributed then $y = w^T \mathbf{x}$ is also normally distributed with the value of $\frac{S_1}{S_2}$ is of the order $\mathcal{O}(1/N)$. Since $\beta_1, \beta_2$ are very close to 0, we can approximate the expression for $v_1$ as below.

$$v_1 \approx \frac{S_1}{S_2}(1 - \beta_2)^2 \frac{1}{\lambda(1-\beta_1)(1-\beta_2)} = \mathcal{O}(1/N\lambda)$$

$\square$

**Importance scores from a standard estimator.**

When $c_1^{(1)} = (\beta_1 u + (1 - \beta_1)v)^T z^{(i)}$, $\quad c_2^{(i)} = (\beta_2 u + (1 - \beta_2)v)^T z^{(i)}$

we can derive the value of the label by their scaled difference as shown below

$$\frac{(1 - \beta_2)c_1 - (1 - \beta_1)c_2}{(1 - \beta_2)\beta_1 - (1 - \beta_1)\beta_2} = \frac{(1 - \beta_2)c_1 - (1 - \beta_1)c_2}{\beta_1 - \beta_2} = u^T z_i = y_i$$

$$\implies \frac{1 - \beta_2}{\beta_1 - \beta_2}c_1 + \frac{1 - \beta_1}{\beta_1 - \beta_2}c_2 = y_i$$

$$\implies v_1 = \frac{1 - \beta_2}{\beta_1 - \beta_2}, v_2 = \frac{1 - \beta_1}{\beta_1 - \beta_2}$$

# F ADDITIONAL EXPERIMENT DETAILS

**List of fruit concepts from Section 4.1.**

```
apple, apricot, avocado, banana, blackberry, blueberry, cantaloupe,
cherry, coconut, cranberry, cucumber, currant, date, dragonfruit,
durian, elderberry, fig, grape, grapefruit, guava, honeydew, kiwi,
lemon, lime, loquat, lychee, mandarin orange, mango, melon, nectarine,
orange, papaya, passion fruit, peach, pear, persimmon, pineapple, plum,
pomegranate, pomelo, prune, quince, raspberry, rhubarb, star fruit,
strawberry, tangerine, tomato, watermelon
```

**List of animal concepts from Section 5.**

```
lion, tiger, giraffe, zebra, monkey, bear, wolf, fox, dog, cat,
horse, cow, pig, sheep, goat, deer, rabbit, raccoon, squirrel, mouse,
rat, snake, crocodile, alligator, turtle, tortoise, lizard,
chameleon, iguana, komodo dragon, frog, toad, turtle, tortoise,
leopard, cheetah, jaguar, hyena, wildebeest, gnu, bison, antelope,
gazelle, gemsbok, oryx, warthog, hippopotamus, rhinoceros, elephant
seal, polar bear, penguin, flamingo, ostrich, emu, cassowary, kiwi,
koala, wombat, platypus, echidna, elephant
```

**Concepts used for *car* and *plane* from Section 5**

```
car: headlights, taillights, turn signals, windshield, windshield vipers,
     bumpers, wheels
plane: wings, landing gear, sky
```

**Scene labels considered in Section 6.**

```
/a/arena/hockey, /a/auto_showroom, /b/bedroom, /c/conference_room, /c/corn_field
 /h/hardware_store, /l/legislative_chamber, /t/tree_farm, /c/coast,
 /p/parking_lot, /p/pasture, /p/patio, /f/farm, /p/playground, /f/field/wild
 /p/playroom, /f/forest_path, /g/garage/indoor
 /g/garage/outdoor, /r/runway, /h/harbor, /h/highway
 /b/beach, /h/home_office, /h/home_theater, /s/slum,
 /b/berth, /s/stable, /b/boat_deck, /b/bow_window/indoor,
/s/street, /s/subway_station/platform, /b/bus_station/indoor, /t/television_room,
 /k/kennel/outdoor, /c/campsite, /l/lawn, /t/tundra, /l/living_room,
 /l/loading_dock, /m/marsh, /w/waiting_room, /c/computer_room,
/w/watering_hole, /y/yard, /n/nursery, /o/office, /d/dining_room, /d/dorm_room,
 /d/driveway
```

## F.1 ADDITION RESULTS FOR SECTION 6

We report also the tau (Wikipedia, 2023b) distance from concept explanations computed by *Simple* as a measure of explanation quality. Kendall Tau is a standard measure for measuring distance between two ranked lists. It does so my computing number of pairs with reversed order between any two lists. Since *Simple* can only estimate the importance of concepts that are correctly annotated in the dataset, we restrict the comparison to only over concepts that are attributed non-zero importance by *Simple*.

| Dataset↓ | TCAV | O-CBM | Y-CBM | U-ACE |
|---|---|---|---|---|
| ADE20K | 0.36 | 0.48 | 0.48 | **0.34** |
| PASCAL | 0.46 | 0.52 | 0.52 | **0.32** |

Table 3: *Quality of explanation comparison.* Kendall Tau Distance between concept importance rankings computed using different explanation methods shown in the first row with ground-truth. The ranking distance is averaged over twenty labels. U-ACE is better than both Y-CBM and O-CBM as well as TCAV despite not having access to ground-truth concept annotations.

# G   ABLATION STUDY

## G.1   UNCERTAINTY OF CONCEPT ACTIVATIONS

As explained in Section 3.1, we estimate the uncertainty on concept activations using a measure on predictability of the concept as shown in Proposition 1. In this section we evaluate the quality of estimated uncertainty and compare with other (potentially simpler) variants of uncertainty estimation. More crisply, we ask the following question.
*Why not estimate uncertainty using any other uncertainty quantification method?*

We conduct our study using the ResNet-18 model pretrained on Places365 and Pascal dataset that were discussed in Section 6. We use human-provided concept annotations to train per-concept linear classifier on the representation layer. We retained only 215 concepts of the total 720 concepts that have at least two positive examples in the dataset. We then evaluated the per-concept linear classifier on a held-out test set to obtain macro-averaged accuracy. The concepts with poor accuracy are the ones that cannot be classified linearly using the representations. Therefore the error rate per concept is the ground-truth for uncertainty that we wish to quantify.

We may now evaluate the goodness of uncertainty: $\epsilon$ of U-ACE by comparing it with ground-truth (error-rate); observe they are both K-dimensional vectors. We report two measures of similarity in Table 4: (1) Cosine-Similarity (Cos-Sim) between $\epsilon$ and error-rate, (2) Jaccard Similarity (JS) (https://en.wikipedia.org/wiki/Jaccard_index) between top-k least uncertain concepts identified using error-rate and $\epsilon$. For any two vectors $u, v$, and their top-k sets: $S_1(u), S_2(v)$, the Cos-Sim and JS are evaluated as follows.

$$\text{Cos-Sim}(u, v) = \frac{u^T v}{\|u\|\|v\|}$$
$$JS(S_1(u), S_2(v)) = \frac{|S_1(u) \cap S_2(v)|}{|S_1(u) \cup S_2(v)|}$$

We will now introduce two other variants of estimating $\epsilon$.
**MC Sampling.** We may simply repeat the estimation procedure several times (denote by S) with different seed and data split to sample multiple concept activation vectors: $\{a_k^{(1)}, a_k^{(2)}, \ldots, a_k^{(S)}\}$ $k \in [1, K]$. We empirically estimate per-concept uncertainty by averaging over examples: $\epsilon^{MC} = \mathbb{E}_{\mathbf{x} \in \mathcal{D}}[\text{std}([\mathbf{x}^T a_k^{(1)}, \mathbf{x}^T a_k^{(2)}, \ldots, \mathbf{x}^T a_k^{(S)}])]$. Where std is the sample standard deviation: $\text{std}(b_1, b_2, \ldots, b_S) = \sqrt{\frac{\sum_s (b_s - \frac{\sum_s b_s}{S})^2}{S-1}}$. We simply repeated the estimation procedure of Oikarinen et al. (2023) that is summarized in Section 3.1 multiple times with different seed and data split to sample different activation vectors.

**Distribution Fit**. Inspired by ProbCBM proposed in Kim et al. (2023a), we estimate $\epsilon$ from the data as a learnable parameter through distribution fitting. We assume normal distrbution of the noise and model the standard deviation as a linear projection of the feature vector. The model is summarized below.

$$g(\mathbf{x})^T g_{text}(T_k) \sim \mathcal{N}(\mu_k(\mathbf{x}), \sigma_k^2(\mathbf{x}))$$
$$\mu_k(\mathbf{x}) = \vec{p_k}^T f(\mathbf{x}), \sigma_k(\mathbf{x}) = \vec{q_k}^T f(\mathbf{x})$$

We obtain the observed score of a concept given an example: $\mathbf{x}$ using the multi-modal model: $g, g_{text}$ and the text description of the $k^{th}$ concept $T_k$. The concept score is modeled to be distributed by a normal distribution whose mean and standard deviation are linear functions of the feature representation of the model-to-be-explained: $f^{[-1]}(\mathbf{x})$. We optimize the value for $[\vec{p_1}, \vec{p_2}, \ldots, \vec{p_K}], [\vec{q_1}, \vec{q_2}, \ldots, \vec{q_K}]$ through gradient descent on the objective $\beta = MLL(\mathcal{D}, g) + \beta \times \mathbb{E}_{\mathcal{D}}\mathbb{E}_k[KL(\mathcal{N}(0, I)\|\mathcal{N}(\mu_k(\mathbf{x}), \sigma_k(\mathbf{x})))]$ very similar to the proposal of Kim et al. (2023a). We picked the best value of $\beta$ and obtained $\epsilon^{DF}$ by averaging over all the examples: $\epsilon^{DF} = \mathbb{E}_{\mathcal{D}}[\sigma_k(\mathbf{x})]$.

**Evaluation of epistemic uncertainty.** We compared the estimate of uncertainty obtained through MC sampling ($\epsilon^{MC}$ with hundred samples) and Distribution Fitting ($\epsilon^{DF}$) with $\epsilon$ of U-ACE in Table 4. We observe that uncertainty obtained using distributional fitting is decent without incurring huge computational cost, however U-ACE produced the highest quality uncertainty at the same or slightly lower computational cost of distributional fitting.

| Method | Cos-Sim | Top-10 | Top-40 | Top-80 |
|---|---|---|---|---|
| MC Sampling | -0.13 | 0 | 0.08 | 0.21 |
| Distribution Fit | 0.06 | 0.11 | 0.19 | 0.31 |
| U-ACE | **0.36** | 0.11 | **0.29** | **0.36** |

Table 4: Evaluation of uncertainties estimated using U-ACE, MC sampling and Distribution Fit (see text for their description). Cos-Sim is the cosine-similarity with ground-truth value of uncertainty. The next three columns show Jaccard similarity between the top-k concepts ranked by ground-truth uncertainty and each of the three methods.

**Evaluation of uncertainty due to ambiguity.** The results so far have confirmed the merits of U-ACE over the other two in modelling the uncertainty due to lack of information. In Figures 7,8,9,10, we present anecdotal evidence that U-ACE is very effective at modelling uncertainty due to ambiguity. In each figure, we compare most (first two columns) and least (last two columns) uncertain images identified by Distribution Fit (in the first row) and U-ACE in the second row.

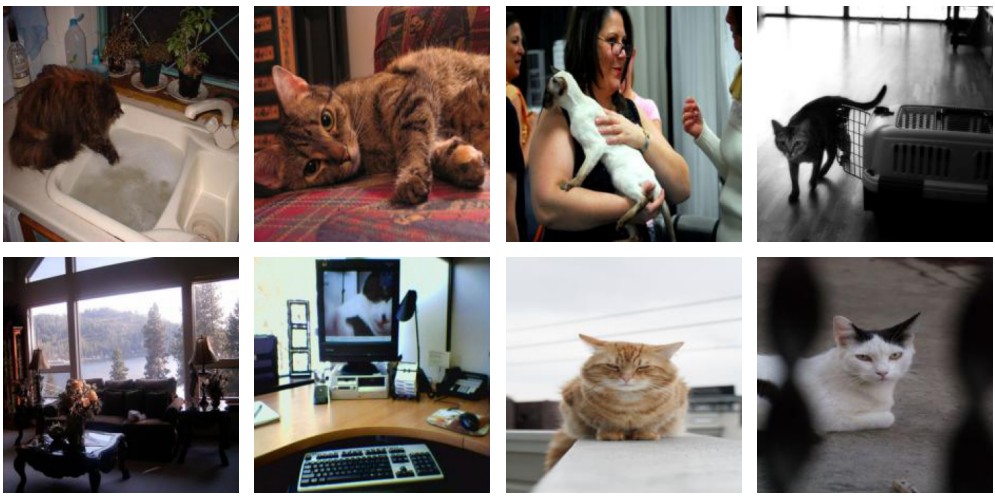

Figure 7: Comparison of ambiguity ranking for **Cat** with Distr. Fit in the top row and U-ACE in the bottom row. Most uncertainty (due to ambiguity) on the left to least uncertainty on the right.

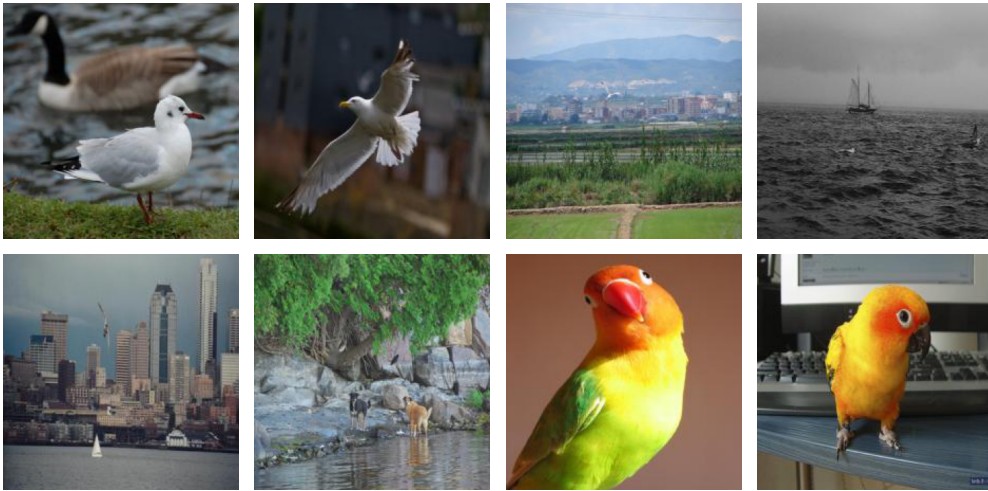

Figure 8: Comparison of ambiguity ranking for **Bird** with Distr. Fit in the top row and U-ACE in the bottom row. Most uncertainty (due to ambiguity) on the left to least uncertainty on the right.

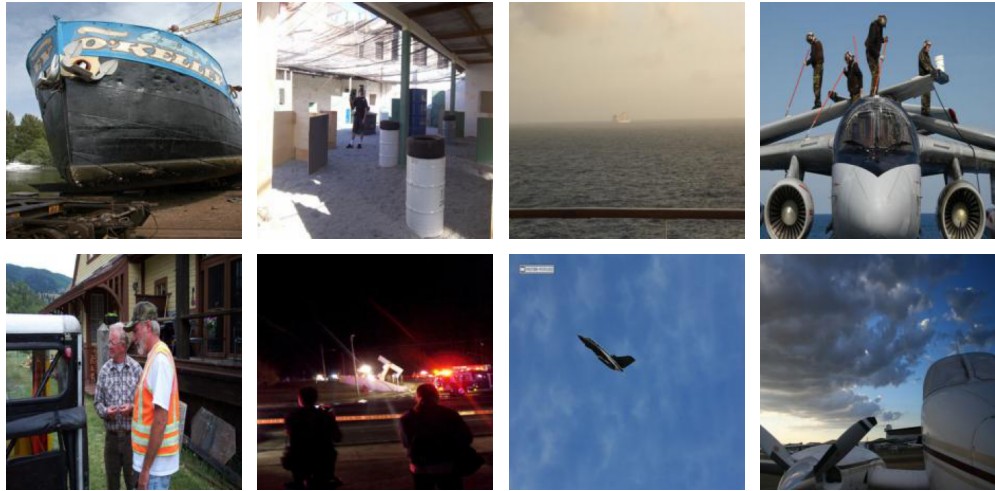

Figure 9: Comparison of ambiguity ranking for **Sky** with Distr. Fit in the top row and U-ACE in the bottom row. Most uncertainty (due to ambiguity) on the left to least uncertainty on the right.

### G.2 BAYESIAN ESTIMATION AND SIGNIFICANCE OF PRIOR

The focus of this section is to motivate the uncertainty-aware prior used by U-ACE. More crisply, the subject of this section is to answer the following question.
*What is the role of prior in U-ACE, and what happens without it?*

We replicate the study on Broden dataset of Section 6 of Table 1 with two new baselines. We replace the linear model estimation of U-ACE described in Section 3 with an out-of-the-box Bayesian Regression estimator available from sklearn[1], which we refer as Bayes Regr. Effectively, Bayes Regr. is different from U-ACE only in the prior. We also compare with the estimation of fitting using Bayes Regr. but when the input is perturbed with the noise estimated by U-ACE. We refer to this baseline as Bayes Regr. with MC.

Table 5 contrasts the two methods that differ majorly only on the choice of prior with U-ACE. We observe a drastic reduction in the quality of explanations by dropping the prior.

---

[1]https://scikit-learn.org/stable/modules/generated/sklearn.linear_model.BayesianRidge.html

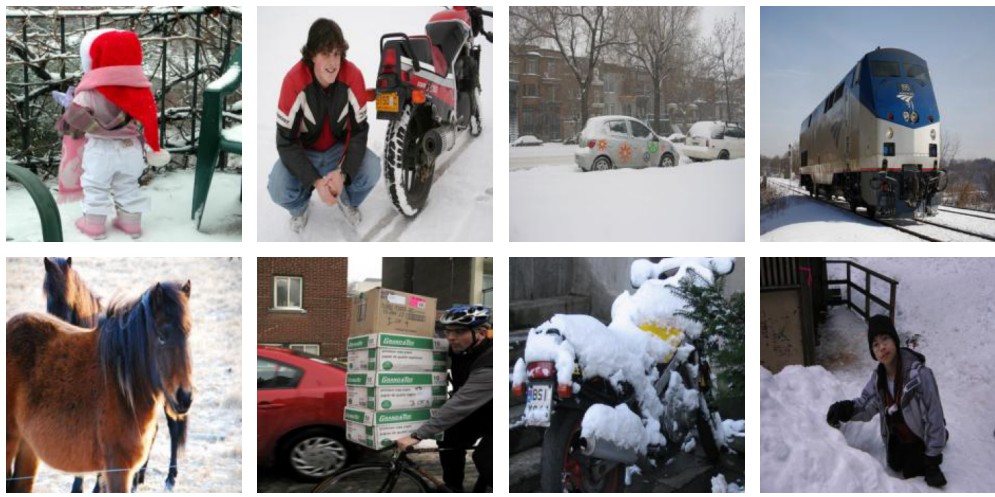

Figure 10: Comparison of ambiguity ranking for **Snow** with Distr. Fit in the top row and U-ACE in the bottom row. Most uncertainty (due to ambiguity) on the left to least uncertainty on the right.

*Can we trivially fix TCAV with a simple uncertainty estimate?*
TCAV is already equipped with a simple uncertainty measurement to distinguish a truly important concept from a random concept. TCAV computes $\vec{m(x)}$ and $\vec{s(x)}$ of concept activations by simply training multiple concept activation vectors. Yet, TCAV estimated explanations are noisy as seen in Table 1 and in the top-10 salient concepts shown below. The poor quality of TCAV explanations despite employing uncertainty (although in a limited capacity) is likely because simple measurement of uncertainty through MC sampling is not the best method for estimating uncertainty as shown in Table 4.

| Dataset | Bayes Regr. | Bayes Regr. with MC | U-ACE |
|---------|-------------|---------------------|-------|
| ADE20K  | 0.39        | 0.43                | **0.09** |
| Pascal  | 0.40        | 0.45                | **0.11** |

Table 5: Significance of prior: quality of explanations severely degrades without the uncertainty-aware prior.

The tables below give a detailed view of the top-10 salient concepts identified using ADE20K for the ResNet-18 scene classification model. The problematic or outlandish concepts are marked in red. We observe that although Bayesian Regr. and Y-CBM are practically the same, the choice of the estimator and sparsity seems to have helped Y-CBM produce (seemingly) higher quality explanations.

Label: **Tree Farm** (ADE20K)

| | |
|---|---|
| TCAV | palm, horse, pane of glass, helicopter, rubbish, cap, boat, organ, tent, footbridge |
| Bayes Regr. with MC sampling | net, merchandise, labyrinth, black, big top, ottoman, chest, pigeonhole, tree, sky |
| Bayes Regr. | oar, forest, pigeonhole, merchandise, sand trap, net, wallpaper, tray, calendar, tree |
| O-CBM | forest, pot, pottedplant, hedge, trestle, sweater, bush, leaf, foliage, coat |
| Y-CBM | field, forest, foliage, elevator, gravestone, hedge, bush, vineyard, covered bridge, baptismal font |
| U-ACE | foliage, forest, grass, field, hedge, covered bridge, tree, leaves, bush, gravestone |

Label: **Coast** (ADE20K)

| | |
|---|---|
| TCAV | shutter, manhole, baby buggy, umbrella, sand, boat, arch, minibike, rubbish, column |
| Bayes Regr. with MC sampling | wineglass, guitar, headlight, chest, jersey, roundabout, witness stand, magazine, folding door, shaft |
| Bayes Regr. | lake, headlight, island, hen, dog, chest, jersey, mosque, shaft, windshield |
| O-CBM | sea, island, lighthouse, cliff, wave, shore, rock, sand, pitted, crystalline |
| Y-CBM | sea, sand, lake, island, runway, cliff, fog bank, clouds, towel rack, pier |
| U-ACE | sea, lake, island, pier, cliff, lighthouse, shore, fog bank, water, sand |

### G.3 EFFECT OF REGULARIZATION STRENGTH ON Y-CBM AND O-CBM

We present the sensitivity analysis for the two strong baselines: Y-CBM and O-CBM in this section.

*How are the hyperparams tuned?*
Hyperparameter tuning is tricky for concept explanations since they lack a ground-truth or validation set. The reported results for Y-CBM and O-CBM in the main paper used the default value of the regularization strength of the corresponding estimator, which is `Lasso`[2] for Y-CBM and `LogisticRegression`[3] for O-CBM. Both the estimators are part of sklearn. We had to reduce the default regularization strength of Y-CBM to $\alpha = 10^{-3}$ so that estimated weights are not all 0. The $\kappa$ of U-ACE is somewhat arbitrarily set to 0.02 on Broden dataset for sparse explanation with non-zero weight for only 20-30% of the concepts.

*Can Y-CBM and O-CBM do much better if we tune the regularization strength?*
We present the results of the two baselines for various values of the regularization strength in Table 6. The table shows quality of explanations in the first two rows for the same setup as Table 1 and also shows the measure of drift in explanations like in Table 2. We tried C=1e-2, 0.1, 1, 10 for O-CBM and $\alpha$ =1e-4, 1e-3, 1e-2, 1e-1 for Y-CBM. We dropped C=1e-2 and $\alpha$ =1e-2, 1e-1 from the table because then the exaplanations were overly sparsified to zero.

We observe from the table that U-ACE still is the best method that that yields high-quality explanation while also being less sensitive to shift in the probe-dataset.

| Dataset | O-CBM | | | Y-CBM | | U-ACE |
|---|---|---|---|---|---|---|
| Regularization strength → | C=0.1 | C=1 | C=10 | $\alpha=10^{-4}$ | $\alpha=10^{-3}$ | $\kappa$=0.02 |
| ADE20K | 0.12 | 0.20 | 0.29 | 0.24 | 0.14 | **0.09** |
| Pascal | 0.11 | 0.25 | 0.35 | 0.27 | 0.13 | **0.11** |
| ADE20K→Pascal | 0.46 | 0.26 | 0.12 | 0.29 | 0.34 | **0.19** |

Table 6: Results on Broden dataset with varying value of regularization strength for O-CBM and Y-CBM. ADE20K and Pascal rows compare the distance between the explanations computed from the ground-truth exactly like Table 1. The last row compares how much the explanations drifted between the datasets exactly like in Table 2. Lower is better everywhere. Observe that U-ACE has high explanation quality while also being relatively more robust to data shift.

## H EXTENSION OF SIMULATION STUDY

**Under-complete concept set**. We now generate concept explanations with concepts set to {*"red or blue", "blue or red", "green or blue", "blue or green"*}. The concept *"red or blue"* is expected to be active for both *red* or *blue* colors, similarly for *"blue or red"* concept. Since all the concepts contain a color from each label, i.e. are active for both the labels, none of them must be useful for prediction. Yet, the importance scores estimated by Y-CBM and O-CBM shown in the Figure 7

---

[2]https://scikit-learn.org/stable/modules/generated/sklearn.linear_model.Lasso.html

[3]https://scikit-learn.org/stable/modules/generated/sklearn.linear_model.LogisticRegression.html

table attribute significant importance. U-ACE avoids this problem as explained in Section 3.2 and attributes almost zero importance.

| Concept | Y-CBM | O-CBM | U-ACE |
|---|---|---|---|
| red or blue | -75.4 | -1.8 | 0.1 |
| blue or red | 21.9 | -1.9 | 0 |
| green or blue | -1.4 | 1.6 | 0 |
| blue or green | -23.1 | 1.6 | 0 |

Table 7: When the concept set is under-complete and contains only nuisance concepts, their estimated importance score must be 0.

# I  EVALUATION USING CUB DATASET

Wah et al. (2011) released a bird dataset called CUB with 11,788 images and 200 bird species. Moreover, each bird image is annotated with one of 312 binary attributes indicating the presence or absence of a bird feature. Koh et al. (2020) popularized an improved version of the dataset that retained only 112 clean attribute annotations. We evaluate using the cleaner dataset released by Koh et al. (2020) owing to their popularity in evaluating CBMs. We train a pretrained ResNet-18 model using the training split of CUB dataset. We then compute the explanation (i.e. saliency of concepts) using the test split. Similar to the evaluation of Section 6, we quantify the quality of explanations using distance from explanations computed using true concept annotations when using *Simple*.

|  | TCAV | O-CBM | Y-CBM | U-ACE |
|---|---|---|---|---|
| Top 3 | **0.38** | 0.43 | 0.46 | **0.43** |
| Top 5 | **0.39** | 0.45 | 0.46 | **0.44** |
| Top 10 | **0.37** | 0.43 | 0.45 | **0.42** |
| Top 20 | **0.36** | 0.40 | 0.41 | **0.39** |

Table 8: Distance of top-k salient concepts computed using *Simple* and different estimation methods shown in the first row (lower the better). TCAV does well overall and U-ACE performs the best among methods without access to concept annotations.

|        | TCAV | O-CBM | Y-CBM | U-ACE |
|--------|------|-------|-------|-------|
| Top 3  | **0.22** | 0.17 | 0.13 | **0.20** |
| Top 5  | **0.5**  | 0.33 | 0.28 | **0.45** |
| Top 10 | **1.68** | 1.18 | 1.02 | **1.34** |
| Top 20 | **5.16** | 4.185 | 3.935 | **4.51** |

Table 9: Average overlap between top-k salient concepts computed using *Simple* and different estimation methods shown in the first row (higher the better). TCAV does well overall and U-ACE performs the best among methods without access to concept annotations.

