# OpenReview forum: "Estimation of Concept Explanations Should be Uncertainty Aware"
_ICLR.cc/2024/Conference — Submitted to ICLR 2024_

### Official Review · Reviewer_6L9H · 2023-10-29

**Soundness:** 1 poor
**Presentation:** 1 poor
**Contribution:** 1 poor
**Rating:** 3
**Confidence:** 4

**Summary:**

The goal of the paper is clearly stated in the title: estimate the uncertainty of concept-based explanations.
The authors present some theories the cornerstone of which is that if the importance of a concept is low, its uncertainty should be high, and conversely.

**Strengths:**

* The idea is indeed an important issue

**Weaknesses:**

* The theory is poorly presented.
* The usage of CLIP seems essential, yet it is not involved in the synthetic experiments.
* The text contains too many typos or missing words.

**Questions:**

Section 2:
* Using L for the number of labels is confusing since l is used for many other things later. Why not M?
* If $v_k$ is defined with respect to a layer $l$, why is there no subscript on $v_k$?
* TYPO: Shouldn't it be $\mathcal{D}_c^{(k)}$ in the expected value instead of $\mathcal{D}_c^{(k)}$?
* The end of the first paragraph of the subsection "Data-effcient concept explanations" is confusing.
* The limitation subsection highlights that such models are very dependent on the probe-dataset, is U-ACE robust to that issue?
* $W_c s(x) = 0 $ means that either $W_c = 0$ or all the $s(x)$ lie on the kernel of $W_c$ which is not impossible. I infer from the text that the second case does not occur: why?
* Again $W_c \epsilon = 0 $ means that either $W_c=0$ or $\epsilon$ is in the kernel of $W_c$. The second case is even more likely than in the previous point. If you don't justify that, you can't unroll your sequence of implications.
* * Since the number of concepts is usually larger than the number of labels, the kernel is often not reduced to zero.
* * Since the sequence of implication is not proven nor properly justified, the rest of the theory builds on something frail.
* Could you be more precise regarding the order of magnitude of $\lambda$? Should it be large or small?
* "prior/posterior _on_ weights": the preposition 'on' is not very clear. In any case, could you give more details about how you obtained Equation 1?

Section 3.1
* What is "cos-sim"?
* What exactly is $\alpha_k$?
* How big is $N$? It is usually pretty large, so how do you optimize something with let's say 10K dimensions?

Proposition 1
* What is $\theta_k$?
* The proof relies heavily on the statement: "If the examples in D are diversely distributed without any systematic bias, $A^T A$ is proportional to the identity matrix, meaning the basis of G and W are effectively the same." Can you justify or prove that? Without the point, the proof is not valid, hence the rest neither.
* * In practice, for $A^T A$ to be diagonal all the $g(x_i)$ needs to be orthogonal to each other. Since N is usually much larger than D or S, this cannot happen.
* * If it would, for $A^T A$ to be proportional to the identity, the $g(x_i)$ need to be of the same norm.
* "an arbitrary new example $x$ that is at an angle of $\theta$ from $w_k$", I guess you mean $g(x)$ is an angle $\theta$ from $w_k$.
* How do you go from $cos(w, g(x)) = cos(\theta)cos(\alpha_k ) ± sin(\theta)sin(\alpha_k )$ to $m(x)=cos(\theta)cos(\alpha_k )$ and $s(x)=sin(\theta)sin(\alpha_k )$ How do you know which summant is $m$ and the other is $s$?

Algorithm 1
* If $Y$ is computed only for label $y$ of the for-loop, it should be indicated.

Page 5, paragraph "Unreliable explanations due to over-complete concept set"
* What is $u_k$ and $\sigma_k$?
* What is a _vanilla_ linear estimator?
* "the probability that at least of the K-1 random concepts is estimated to be more important than the relevant concept is $1-\prod \Phi( \dfrac{||u_k||}{\sigma_k ||w||} )$". How do you get this formula?
* * "the CDF of standard normal", some words are missing here.

Proposition 3
* $v_1,v_2 = \mathcal{O}(1/N\lambda)$: this notation is used for limits, not for upper bounds. Even if, you need to show that $||v_1||$ is upper-bounded by $1/N \lambda$.

Section 4
* "pretrained CLIP model that is publicly available for download." Can you provide a link or reference?

Section 4.1
* I understood from Section 3, that U-ACE is built upon a text embedding. How is it learned here?

---

> ### Author Response · Authors · 2023-11-19
> **Response (1/2)**
>
> We thank the reviewer for their valuable time and effort. We are glad that the reviewer appreciates the need to model uncertainty in concept explanations.
>
> > The usage of CLIP seems essential, yet it is not involved in the synthetic experiments.
>
> Sorry for any confusion. We would like to point out that U-ACE uses CLIP to encode concepts in all the experiments including synthetic. As described in Section 3.1 and Proposition 1, all our experiments, including synthetic experiments, used text descriptions to encode concepts using the multi-modal CLIP model.
>
> > TYPO: Shouldn't it be $\mathcal{D}_c^{(k)}$ in the expected value instead of $\mathcal{D}_c^{(k)}$?
>
> Sorry, looks like your question has a typing mistake, we believe you may be asking $\mathcal{D}_c^{(k)}$ instead of $\mathcal{D}$. If so, we do not see any typo on that line in Algorithm 1. We estimate the expectation over the probe-dataset ($\mathcal{D}$) and not over the concept dataset ($\mathcal{D}_c^{(k)}$). U-ACE assumes $\mathcal{D}_c^{(k)}$ is not available (please see inputs of U-ACE shown in Algorithm 1). $\mathcal{D}_c^{(k)}$ were introduced only to explain how previous concept explanation methods such as TCAV estimate concept activation vectors.
>
> > The limitation subsection highlights that such models are very dependent on the probe-dataset, is U-ACE robust to that issue?
>
> Yes, we demonstrated the relative robustness of U-ACE to probe-dataset in Table 2 (“Dataset shift” of Section 4.3) and Figure 3 (“Unreliability due to dataset shift.” in Section 4.1).
>
> > Again $W_c\epsilon$ means that either $W_c=0$ or $\epsilon$ is in the kernel of $W_c$. … I infer from the text that the second case does not occur: why?
>
> Sorry, there could be some misunderstanding. We did not assume anything about the kernel or the matrix. We estimated $W_c$ that keeps the value of $W_c\epsilon$ low while also approximating the logits well (please see Alg. 1). U-ACE estimated $W_c=0$ only when all the concepts are irrelevant (please see Appendix F.1 and Proposition 3 for an example). In every other case, the estimated $W_c\neq 0$, which is why we see meaningful explanations such as in Figure 6.
>
> > Could you be more precise regarding the order of magnitude of $\lambda$?
>
> If $\lambda$ is very low the constraint $W_c\epsilon\approx 0$ is satisfied but the approximation error can be large. Whereas if the $\lambda$ is very high then the unconstrained regression may lead to better approximation but model is not unaware of uncertainty, i.e. $\|W_c\epsilon\|$ is likely large.
> $\lambda$ therefore must be neither very small nor very large. We found the best value of $\lambda$ through maximum likelihood estimation (MLE) on training data as described in Appendix B. Therefore the best value of $\lambda$ varied per dataset and was set automatically by MLE.
>
> > "prior/posterior on weights": the preposition 'on' is not very clear. In any case, could you give more details about how you obtained Equation 1?
>
> By prior/posterior on weights, we meant $\Pr(\vec{w})$ and $\Pr(\vec{w}\mid \mathcal{D})$ (where $\vec{w}$ according to the notation on page 3 is a row of $W_c$). Please see Slide 10 of https://www.utstat.toronto.edu/~rsalakhu/sta4273/notes/Lecture2.pdf#page=10 for how Equation (1) is derived, note that $m_0=0, S_0^{-1}=\lambda^{-1}\text{diag}(\epsilon\epsilon^T)$, $\Phi=C_X$, Y as defined above Equation 1. We also added the full derivation to Appendix A.1 for the sake of completeness.
>
> > What is "cos-sim"?
>
> Thanks for pointing it out, we made it more explicit. We used “cos-sim” as a short form for cosine similarity (https://en.wikipedia.org/wiki/Cosine_similarity).
>
> > What exactly is $\alpha_k$
>
> The concept specific value $\alpha_k$ is a measure of uncertainty due to lack of knowledge  and due to ambiguity. We added text to Section 3.1 to make the definition clear.
>
> > How big is N? It is usually pretty large, so how do you optimize something with let's say 10K dimensions?
>
> N is the number of examples in the probe-dataset and can be very large. But in our experiments, the maximum value of N was only about 10,000 (on the Broden dataset). We pre-computed $e(w_k, g, \mathcal{D})$ and computed $e(v, f, \mathcal{D})$ at every step, and fully loaded both the vectors of dimension N into memory. Thanks for raising this fair point, we may need some special handling for very very large N. We will comment on the same in the revised version.
>
> > What is $\theta_k$?
>
> $cos(\theta_k)$ as defined in Proposition 1 is the cosine-similarity between the embedding of the image (x) and the embedding of the concept k ($T_k$).

---

> ### Author Response · Authors · 2023-11-19
> **Response (2/2)**
>
> > The proof relies heavily on the statement: "If the examples in D are diversely distributed without any systematic bias, is proportional to the identity matrix, meaning the basis of G and W are effectively the same." Can you justify or prove that? Without the point, the proof is not valid, hence the rest neither.
>
> There may have been some confusion. $A^TA$ is of size $S\times S$ where S is the rank of G. For $A^TA$ to be proportional to an identity matrix, we only require that the examples in the probe-dataset are such that $[g(x_1), g(x_2), …]$ are uniformly spread over the subspace of dimension S. We demonstrate this with an example.
> ```
> def makeA(N, S):
> 	return np.array([[np.random.uniform(-1, 1) for s in range(S)] for n in range(N)])
> def makeA2(N, S):
> 	return np.array([[np.random.uniform(-s-1, s+1) for s in range(S)] for n in range(N)])
> >>> A = makeA(1000, 2)
> >>> A.transpose() @ A
> [[334.4,   8.6],
> [  8.6, 338.7]])
> >>> A = makeA2(1000, 2)
> >>> A.transpose() @ A
> [[ 341.4,   24.3],
>  [  24.3, 1277.0]]
> ```
>
> For the second matrix from makeA2, the first dimension is sampled from Unif[-1, 1] while the second from Unif[-2, 2]. Since all the dimensions are not identically distributed, it has systematic biases which prevented $A^TA$ (of makeA2) from being proportional to a diagonal matrix.
>
> We understand that our assumption is not guaranteed to hold since it depends on the diversity of the probe-dataset. Yet, the estimates were found to be empirically very effective as observed implicitly from the high quality of explanations. Moreover, direct evaluation of uncertainty, presented in Appendix G.1, further confirm the empirical effectiveness of U-ACE.
>
> > How do you go from $cos(\theta\pm \alpha_k)$ to … How do you know which summant is m(x) and the other is s(x)?
>
> By definition of m(x), s(x), concept activations lie in the range of $m(x)\pm s(x)$. The proof of Proposition 1 finds the range of concept activations as $cos(\theta)cos(\alpha_k)\pm sin(\theta)sin(\alpha_k)$. The result (of $m(x)=cos(\theta)cos(\alpha_k), s(x)=sin(\theta)sin(\alpha_k)$) follows from directly reading off from the estimated range.
>
> > What is $u_k$ and $\sigma_k$?
>
> They are the mean and scale respectively of the normal distribution from which the concept activation vectors (w_k) are sampled.
>
> > What is a vanilla linear estimator?
>
> We meant ordinary least squares (OLS) by vanilla linear estimator. We corrected the phrase in the latest draft to avoid ambiguity.
>
> > "the probability that at least of the K-1 random concepts is estimated to be more important than the relevant concept is ". How do you get this formula?
>
> We modified the text to introduce Corollary 1 and presented the derivation in the proof of Corollary 1. Please see the revised draft.
>
> > "the CDF of standard normal", some words are missing here.
>
> We meant $\Phi$ is the Cumulative Distribution Function of standard normal distribution.
>
> > "pretrained CLIP model that is publicly available for download." Can you provide a link or reference?
>
> Sure, https://github.com/openai/CLIP/. We also added it in the latest draft.
>
> > I understood from Section 3, that U-ACE is built upon a text embedding. How is it learned here?
>
> U-ACE uses text description of a concept to encode a concept through a pretrained multi-modal model such as CLIP. But U-ACE itself does not train any text embeddings.
>
> > If Y is computed only for label y of the for-loop, it should be indicated.
>
> Thanks for noticing. We corrected it in the latest draft.
>
> > If $v_k$ is defined with respect to a layer $l$, why is there no subscript on $v_k$?
>
> Thanks for your suggestions. We fixed this in the revision.

---

### Official Review · Reviewer_eXDK · 2023-10-31

**Soundness:** 2 fair
**Presentation:** 2 fair
**Contribution:** 3 good
**Rating:** 6
**Confidence:** 3

**Summary:**

This paper identifies that concept explanation methods are unstable along both concepts choice and dataset choice. The papers discusses how this instability is due to improper noise modeling. To overcome these issues, they propose an uncertainty aware concept explanation method. Through both theoretical and empirical analysis they generally find the concept explanations lead to more stable concept based explanations.

**Strengths:**

- The idea is quite interesting. Concept based explanations are generally known to suffer from key issues related to instability. These issues sometimes affect their use more generally, and this paper does a good job of identifying / offering a potential solution.
- The results concerning increasing concept set vs. introduced noise are quite interesting + methods discussed in 3.1 to help overcome these issues are quite compelling
- The experimental results in 4.2 as irrelevant concepts are added are also quite interesting and indicate the usefulness of the method to overcome noise in concepts -- I think this could be quite a useful method to help overcome these issues.

**Weaknesses:**

- In general, the most significant weakness of this paper was presentation and clarity. The results are quite interesting and relevant to work on concept based explanations. Nevertheless, the paper was quite hard to follow in places and could do with a bit of work to more clearly express the main takeaways and experimental evaluations. In the current form, the paper is quite hard to follow.
- The presentation of the experimental results could be improved. In 4.1, the "Unreliability due to misspecified concept set" subsection feels a bit odd in the flow of the paper, because it serves as a pointer to the appendix, perhaps it would be better if this is incorporated into another paragraph to keep the flow of the paper clearer.

**Questions:**

- Could you contextualize the significant of proposition 3? It understandable that its important this is the case (namely, that near zero importance scores occur), but I would be interested to understand whether this is expected to be the case or not with other concept explanation baselines?

---

> ### Author Response · Authors · 2023-11-19
>
> We are grateful for the reviewer’s thoughtful and passionate feedback. We are very glad that the reviewer found our idea interesting, problem formulation practical, methods compelling/useful and evaluation interesting/convincing.
>
> > … could do with a bit of work to more clearly express the main takeaways and experimental evaluations…
> > The presentation of the experimental results could be improved. In 4.1, the "Unreliability due to misspecified concept set" subsection…
>
> We thank the reviewer for their suggestions. We have rewritten the hard-to-parse sections, filled in missing details and presented all the derivations in the Appendix (Please see general response). We also acted on the suggestion made by the reviewer for Section 4.1.
>
> > Could you contextualize the significant of proposition 3? …
>
> Thanks for your question. The near-zero importance, i.e. the value of importance score being zero for irrelevant concepts, applies only to the baselines that fit a linear model on the concept activations to estimate the explanation, which is the standard linear estimator baseline considered in Proposition 3. The concept importance score may have a different interpretation for other baselines. For instance in TCAV, any importance score that is not statistically greater than 0.5 is equivalent to zero and is an irrelevant concept.  For the ease of analysis, we only considered explanations through linear estimator baselines, which include our two strong baselines: Y-CBM, O-CBM, U-ACE.
>
> Please consider the following response if the question is instead about the need for assigning zero-importance or the need for handling concept sets with no relevant concepts.
>
> Proposition 3 takes the under-complete concept set to the extreme case when all the concepts in the concept set are irrelevant. Appendix F presents an example. In practice, an irrelevant concept set may arise when the human guesses the task-relevant concepts wrong, which is not unusual. Some examples reported in the past are: (a) [1] reported that a standard model (depending on the training data) was found to exploit the green background for recognising cows and yellow background for camels. A user may start cow or camel related concepts and should be able to see they are low importance scores for them all. (b) [2], [3] found that models trained to recognize lung related disease were not even focusing on lung related features. (c) Similar to (b), [4] reported that a model trained to classify skin lesions was found to rely on any lesion related features but instead relied on a special scale that watermarked tumorous lesions.
>
> In all the above scenarios, the user may start with their guess (lung/chest related concepts in (b) and lesion-related concepts for (c)) and should find that the model is not relying on any of the concepts that they considered are relevant.
>
> We are happy to answer any further questions. Thanks again for your careful assessment.
>
> References.
> 1. Beery, Sara, Grant Van Horn, and Pietro Perona. "Recognition in terra incognita." Proceedings of the European conference on computer vision (ECCV). 2018.
> 2. DeGrave, Alex J., Joseph D. Janizek, and Su-In Lee. "AI for radiographic COVID-19 detection selects shortcuts over signal." Nature Machine Intelligence 3.7 (2021): 610-619.
> 3. Zech, John R., et al. "Variable generalization performance of a deep learning model to detect pneumonia in chest radiographs: a cross-sectional study." PLoS medicine 15.11 (2018): e1002683.
> 4. Narla, Akhila, et al. "Automated classification of skin lesions: from pixels to practice." Journal of Investigative Dermatology 138.10 (2018): 2108-2110.

---

### Official Review · Reviewer_xCUE · 2023-11-01

**Soundness:** 2 fair
**Presentation:** 1 poor
**Contribution:** 2 fair
**Rating:** 5
**Confidence:** 4

**Summary:**

This paper proposes a method for estimating concept importance in models in an uncertainty-aware manner. The method builds upon existing work on representation space registration between CLIP models and other deep learning models, providing an approach for unsupervised concept detection. A predefined set of concepts is given, then this approach estimates latent space directions for each concept using a probe set (special dataset), which can be used to compute a concept-activation score for each concept using a given image as well as an uncertainty estimate for that concept-activation score. Once the concept scores and their uncertainties are estimated, a sparse linear model is fit to the data (predicting actual model outputs from class activation scores as inputs) with a prior that encourages the linear model to be robust to concept noise. Experiments assess the ability of the method to avoid attributing importance to unused features, attribute importance in proportion to a concept’s ground-truth known importance, and lastly to produce concept attributions that are close to those from a linear model fit to a model using human concept data (predicting model outputs from human concept annotations over images).

**Strengths:**

- Very Important: The paper addresses two clear and important problems with current concept attribution methods: (1) the need for supervised data in concept estimation and (2) that current methods generally do not include uncertainty estimates for concept vectors (more on uncertainity of concept attributions later).
- Important: The paper carries out a number of experiments that support the method’s ability to pick up on important concepts to model predictions. Beyond simulations with ground-truth concept importance, the method also achieves a better fit against a pseudo-ground-truth “simple” model with access to ground truth annotations when run over a more realistic vision model on real image scenes, relative to methods like TCAV.

**Weaknesses:**

- Very Important: This paper attempts to address two topics at once, and its novelty is severely undercut by existing work in both directions. At least, I say it attempts to address two topics because that is how it is pitched and evaluated, although the technical novelty of the paper is to incorporate uncertainty into concept estimation (and it only uses existing approaches to *unsupervised* concept estimation). Anyway, the paper comes on the heels of a number of works on unsupervised concept detection, including https://openreview.net/pdf?id=iOlYmD1PtC8, https://arxiv.org/pdf/2304.09707.pdf, and https://arxiv.org/pdf/2309.08600.pdf. I believe all of these works appeared within the last six months, so they are concurrent work. However, perhaps more importantly, the unsupervised concept detection in this paper uses strictly existing methods based on CLIP, so there is not novelty on the supervision side. On the concept estimation side, there is older work on the subject, e.g. this 2021 paper https://proceedings.neurips.cc/paper_files/paper/2021/file/4e246a381baf2ce038b3b0f82c7d6fb4-Paper.pdf, in addition to the concurrent 2023 citation in the submission (Probabilistic Concept Bottleneck Models. The 2021 Slack et al paper focuses on linear feature attribution wrt input features rather than concept vectors, but rest of the subject is the same and this paper also introduces a Bayesian model for handling uncertainty in the attribution. Or if this problem setting is not too similar to the one in this paper, what is the difficulty in uncertainty for the vector v_k in Sec. 3.1, which is the concept vector, using any number of standard uncertainty measurements (whether frequentist or Bayesian)?
- Very Important: Core details of the paper are not clear. How is s(x) computed? What is sin(x)? I know *cos*(.) is defined, but the italicized *sin*(.) is never defined? I don’t take it to literally be the sine function. This is the centerpiece of uncertainty estimation, and I really could not tell where the uncertainty was supposed to come from.
- Important: The prior for the regression model is not well motivated. First, since the regression weights are filtered with a threshold for zeroing small weights, it’s clear that a spike-and-slab prior or another sparsity-encouraging prior would have been more appropriate. But the paper aims to encourage weights to be orthogonal to a noise vector. What is this noise vector? It’s a *mean confidence interval vector* judging by Fig. 1 which shows activations as m(x) +/- s(x). This implies that the weights are supposed to be orthogonal to the upper bound of a per-data-point confidence interval, averaged across the data. Maybe I’m misunderstanding something, but at the moment, I have no idea why this would be a good prior for the model.
- Important: Fair comparison with baselines: How was Y-CBM tuned, and what exactly is the final different with U-ACE? Is it just the prior? Is the technical contribution here mainly the prior? How would an L1 or another sparsity inducing prior affect Y-CBM performance in Table 1?
- Of some importance: It doesn’t make sense to ask what the importance of green is when there are no green images in the probe set (Fig3 middle) or what the importance of red is when the images are all green. Just because CLIP registration enables one to ask this question does not make it meaningful, and a practicioner should notice that their probe set contains no example of green before trying to estimate the importance of green. The same applies for the “fruit” example. The quality of concept estimation is heavily bottlenecked by probe set quality, as noted in this paper, but it has to be the practitioner’s responsibility at some point to check the quality and diversity of the probe set before trying to estimate concept importance for concepts that do not appear at all in the set.

**Questions:**

- If the method uses the last hidden layer, isn’t the method basically replacing the final hidden state with an estimated concept feature vector and replacing the final linear layer with a learned linear layer that tries to approximate the original model’s outputs? This is odd because if the concept vectors are directions in the model latent space, then one would think that a concept attribution explanation is explainable merely by computing the projection of the model’s linear layer with the concept vector, and there is no need to recreate a feature vector m(x) for each datapoint (which uses the learned concept vector v only in small part) and learn a new linear model to estimate concept importance.
- So what is the source of uncertainty that U-ACE accounts for? Data uncertainty? Model uncertainty for the deep learned model? If it’s data uncertainity, why can’t TCAV be equipped with a simple data uncertainty measurement? If the TCAV concept attribution score is a binary proportion, then a binomial probability confidence interval would be very easy to use for it, and much more intuitive than s(x).
- “Simple estimates explanation using concept annotations and therefore its explanation must
be the closest to the ground-truth” — What would the ground-truth look like for the experiment in 4.3?
- typo: UNCERTAINITY
- typo: while other methods the importance
- “Since the co-occurrence probability of U with car class goes from 1, 0.5 to 0, we expect the importance score of U should change from positive to negative as we move right”  — It doesn’t change to negative in the graph, although it does get lower

---

> ### Author Response · Authors · 2023-11-19
> **Response (1/2)**
>
> We thank the reviewer for their passionate and thoughtful assessment. We are glad that the reviewer found our motivation and contributions clear, and our evaluation convincing.  We address their concerns below and are very happy to engage further.
> > This paper attempts to address two topics at once, and its novelty is severely undercut by existing work in both directions. … Anyway, the paper comes on the heels of a number of works on unsupervised concept detection, including… However, … this paper uses … CLIP, ... not novelty ….
>
> Sorry, there may have been some misinterpretation. We would like to clarify that U-ACE assumes that the concepts are supplied and we did not propose a method for concept discovery (although it is an important problem in itself). As explained in Section 2, the list of concepts $\mathcal{C}=\{c_1, c_2, \dots, c_K\}$ are supplied along with their text descriptions: $\{T_1, T_2,\dots, T_K\}$.
>
> > On the concept estimation side, there is older work… what is the difficulty in uncertainty for the vector v_k in Sec. 3.1, which is the concept vector, using any number of standard uncertainty measurements (whether frequentist or Bayesian)?
>
> Thanks for raising this issue.  To further motivate the need and effectiveness of uncertainty estimation of U-ACE (described in Section 3.1), we contrasted U-ACE with uncertainty estimated using MC sampling or distributional parameter fitting (similar to ProbCBM as the reviewer suggested). The results can be found in Appendix G.1 of the revised draft. We find that U-ACE estimated uncertainty is high-quality as well as compute-efficient. Thanks for pointing to Slack et.al. work, although their methods are not trivially applicable to our problem we will mention the paper in the related work and highlight the common intent.
>
> > …How is s(x) computed? What is sin(x)? … could not tell where the uncertainty was supposed to come from.
>
>
> sin(.) is in fact the sine function, we will make it explicit. Two major sources of uncertainty in concept activations that are modeled by U-ACE are: (1) epistemic uncertainty arising from lack of information about the concept in the representation layer of the model-to-be-explained, (2) data uncertainty arising from ambiguity likely because the concept is not clearly visible (see Appendix G.1 for some examples). We demonstrate the merits of U-ACE in accounting both the uncertainties in Appendix G.1. We also added passages to Section 3 to clarify the sources of uncertainty.
>
> > … sparsity-encouraging prior would have been more appropriate. But the paper aims to encourage weights to be orthogonal to a noise vector. What is this noise vector? … I have no idea why this would be a good prior for the model.
>
> Sorry for any confusion. We have two motives for the linear estimator $W_c$ (which is returned as the explanation): (1) should be robust to error in concept activations, or in other words, the linear estimator should not rely on concepts with large noise/uncertainty (3rd para of Section 3), (2) should be sparse for easy interpretation. We enforce (1) by imposing a prior such that the samples are almost orthogonal to the noise vector. Please see Equation 1 and text above it for more details about the noise vector ($\epsilon$: a vector of size K: the number of concepts) and justification on why it is a good prior for enforcing the requirement (1) above. Our prior is not designed to enforce sparsity (2); we sparsify weights post-hoc as explained towards the end of Section 3 and before Section 3.1. Further in Appendix G.2, we evaluated the role of prior in producing quality explanations.
>
> > Fair comparison with baselines: How was Y-CBM tuned, and what exactly is the final different with U-ACE? … Is the technical contribution here mainly the prior? How would an L1 or another sparsity inducing prior affect Y-CBM performance in Table 1?
>
>
> Major difference between Y-CBM and U-ACE is that U-ACE estimates the linear weights using Bayesian regression with prior as described in Section 3 while Y-CBM estimates the same using lasso regression. Since the prior on U-ACE’s linear weights is a function of uncertainty in concept activations, it is aware of uncertainty.
>
> Estimation of explanations using Bayesian regression with an uncertainty-aware prior is one the crucial contributions of the paper. We wish to highlight that the contribution of our work also lies in demonstrating the role of uncertainty in concept explanations, and in estimation of uncertainty in concept activations.
>
> To address your concerns more concretely, we did the following: (a) Appendix G.3 presents results for Y-CBM and O-CBM for varying values of the regularization strength, and still perform worse than U-ACE (without any hyperparam tuning), (b) Appendix G.2 contains comparison with Bayesian regression without the uncertainty aware prior (of U-ACE). The experiments paint a clear importance of the U-ACE’s prior.

---

> ### Author Response · Authors · 2023-11-19
> **Response (2/2)**
>
> > …It doesn’t make sense to ask what the importance of green is when there are no green images in the probe set (Fig3 middle) or what the importance of red is when the images are all green. … a practicioner should notice that their probe set contains no example of green before ….
>
> Thanks for your question. We agree that choosing probe-datasets carefully can solve the problem partially as was also suggested by Ramaswamy et.al.: “The choice of the probe dataset has a profound impact on explanations…. Hence, probe datasets should be chosen with caution.” However, when the number of labels, concepts and examples in the probe dataset are in the order of thousands, manually checking for presence of each concept with each label over all the examples can be impractical. Which is why we believe (among other reasons) that a concept explanation must be equipped with confidence scores on their estimate of importance, which insulates from misinterpreting importance scores of missing concepts. Our simulated study with the colored dataset is inspired by the difficulty in carefully picking the probe-dataset or concept set when operating at scale.
>
> > …because if the concept vectors are directions in the model latent space, then one would think that a concept attribution explanation is explainable merely by computing the projection of the model’s linear layer with the concept vector… learn a new linear model to estimate concept importance.
>
> True that “then one would think that a concept attribution explanation is explainable merely by computing the projection of the model’s linear layer with the concept vector”, but such a point estimate is bound to be noisy. m(x) is simply the mean over many such projections (using several samples of concept activation vector).
>
> > So what is the source of uncertainty that U-ACE accounts for?... why can’t TCAV be equipped with a simple data uncertainty measurement? …
>
> We answered the sources of uncertainty in our response above.
>
> TCAV is already equipped with a simple uncertainty measurement to distinguish a truly important concept from a random concept. TCAV computes m(x) and s(x) of concept activations by simply training multiple concept activation vectors. Yet, TCAV estimated explanations are noisy as seen in Table 1 likely because simple measurement of uncertainty through MC sampling may fail to capture epistemic uncertainty as can be observed from the results of Appendix G.1.
>
> > “Simple estimates explanation using concept annotations and therefore its explanation must be the closest to the ground-truth” — What would the ground-truth look like for the experiment in 4.3?
>
> We noticed that concept annotations do not mark all the concepts present in an example, which is understandable given the number of concepts: 720. Since “Simple” is trained on ground-truth concept annotations, it may miss attributing importance to the concepts that are systematically missed by an annotator. We expect therefore that the most salient concepts identified by “Simple” are a subset of truly salient concepts. Nevertheless, “Simple” trained on concept annotations is the closest explanation to ground-truth, which we use to evaluate other methods.
>
> > typo: UNCERTAINITY
> > typo: while other methods the importance
>
> Thank you, we fixed these in the revision.
>
> > “Since the co-occurrence probability of U with car class goes from 1, 0.5 to 0, we expect the importance score of U should change from positive to negative as we move right” — It doesn’t change to negative in the graph, although it does get lower
>
> Thanks for bringing it up. The importance score does go negative but since we visualized in Figure 5 the rank-normalised concept importance scores (as explained on page 6), it can only be positive. We fixed the text accordingly.
>
> We are happy to engage in further discussion to clarify any unresolved concerns. Thanks again for your detailed review.

---

> > ### Comment · Reviewer_xCUE · 2023-11-21
> > **Reply to authors (part 1)**
> >
> > Thanks for the reply! Comments below.
> >
> > > We would like to clarify that U-ACE assumes that the concepts are supplied and we did not propose a method for concept discovery (although it is an important problem in itself).
> >
> > Thanks, the parts that were confusing to me are like this part: "Our objective is to generate reliable concept explanations without requiring datasets with concept annotations.” Really, you mean your objective is to generate good uncertainty measurements for concept explanations, and this is done in a setting that replaces probe datasets with CLIP latent space registration.
> >
> > > The results can be found in Appendix G.1 of the revised draft. We find that U-ACE estimated uncertainty is high-quality as well as compute-efficient.
> >
> > Ok so, the idea is that a concept’s uncertainty can be measured by this per-concept linear probe over the dataset. And if you’re looking for a metric that correlates well with that per-concept linear probe error date, U-ACE’s epsilon does this well and other measurements do not do as well (like MC Sampling, which I think measures the standard deviation of the inner product between the typical x embedding and the concept vector for the concept x is supposed to exhibit). I agree that the error rate, in the presence of a large amount of examples, is a measure of aleatoric/data uncertainty, so I suppose this is a valid evaluation for U-ACE, but see below.
> >
> > > Our objective is to estimate m(x), s(x) such that the true concept activation value is in the range:m(x) ±s(x).
> >
> > This makes sense. Then, the traditional way to measure this is via coverage, not via the error-rate correlation analysis of the appendix G.1. You want calibrated coverage — 95% confidence intervals should cover m(x) 95% of the time. Then coverage can be compared to other measurements like MC Sampling and ProbCBM.
> >
> > > If the model-to-be-explained (f) uses the k th concept for label prediction, the information about the concept is encoded in f and we get a good fit, i.e. cos(αk) ≈ 1, and a small error on concept activations
> >
> > What does this mean? Why does noise go down if the model “uses” the k-th concept for prediction?
> >
> > > sin(.) is in fact the sine function, we will make it explicit.
> >
> > Thanks, I see where the sin(.) appears in the appendix.
> >
> > > the linear estimator (which is returned as the explanation): (1) should be robust to error in concept activations, or in other words, the linear estimator should not rely on concepts with large noise/uncertainty (3rd para of Section 3), (2) should be sparse for easy interpretation
> >
> > Thanks for separating these goals. General question: what should we do if we’re doing model selection on two different linear estimators, and the one that relies on a particular very noisy/fuzzy concept gets a much lower loss against the original model’s predictions than the one that doesn’t rely on this noisy/fuzzy concept. Why not use the estimator that’s a better fit/more faithful to the blackbox model, and admit that it’s relying on a feature that we don’t exactly know the meaning of? I suppose that, empirically, this doesn’t seem to happen, based on results in Sec. 6 and G.3.
> >
> > > We enforce (1) by imposing a prior such that the samples are almost orthogonal to the noise vector.
> >
> > Thanks, I thought about this more and it almost makes sense, but I still feel that there is a flaw. It’s written that, simplifying notation a bit, the objective is for W(m+s) = Wm. But the noise doesn’t work like that. It’s not the case that the possible noise vectors are s and -s. The noise in each element should vary independent, unless a covariance structure is specified. I agree with the stated motivation in the paper: “the linear estimator should not rely on concepts with large noise/uncertainty.” But I don’t think that’s what the math represents. Consider that, the proposed objective of w^Ts=0, could be satisfied for class i by placing positive weight w1 on s_1 and  negative weight w2 on s_2, such that the inner product is 0. This means placing positive weight on m1 and negative weight on m2 (first two elements of m), and can occur irrespective of their uncertainty s.
> >
> > So, what kind of Bayesian prior puts a weight inversely proportional to variance/uncertainty in the feature? It seems that such a prior is really what is preferred, and not the one proposed. However, given that the empirical results in this paper are positive (e.g. the proposed method seems to outperform baselines), I am not exactly sure how to factor in my concerns with the theoretical justification.
> >
> > > Our prior is not designed to enforce sparsity (2); we sparsify weights post-hoc
> >
> > Ok, ultimately this is fine.

---

> > > ### Comment · Reviewer_xCUE · 2023-11-21
> > > **Reply to authors (part 2)**
> > >
> > > > Appendix G.3 presents results for Y-CBM and O-CBM for varying values of the regularization strength, and still perform worse than U-ACE (without any hyperparam tuning), (b) Appendix G.2 contains comparison with Bayesian regression without the uncertainty aware prior (of U-ACE)
> > >
> > > Thanks for adding these. They help clarify the contribution of the paper. It does seem that the proposed method assigns importance scores to concepts that are more similar to “ground-truth” importance scores — this is an added benefit of incorporating uncertainty into the modeling approach.
> > >
> > > I wonder if you could also add in more traditional faithfulness metrics from the XAI literature for evaluating linear explanations like % agreement with the blackbox model (rather than just this the one metric of similarity to the top-20 salient feature’s weights for a pseudo-ground-truth linear estimator).
> > >
> > > > However, when the number of labels, concepts and examples in the probe dataset are in the order of thousands, manually checking for presence of each concept with each label over all the examples can be impractical. Which is why we believe (among other reasons) that a concept explanation must be equipped with confidence scores on their estimate of importance, which insulates from misinterpreting importance scores of missing concepts
> > >
> > > This is a good point and I agree with this.
> > >
> > > > TCAV is already equipped with a simple uncertainty measurement to distinguish a truly important concept from a random concept. TCAV computes m(x) and s(x) of concept activations by simply training multiple concept activation vectors. Yet, TCAV estimated explanations are noisy as seen in Table 1 likely because simple measurement of uncertainty through MC sampling may fail to capture epistemic uncertainty as can be observed from the results of Appendix G.1.
> > >
> > > I see, good point.
> > >
> > > - - - - -
> > >
> > > Overall, I feel that the paper makes some empirical progress on uncertainty estimation for concept explanations, and the author response answered many of my questions about the paper. However, I have concerns about the clarity of presentation and theoretical motivation in the current draft. I think the paper could be rewritten to make the presentation much simpler and more effective, focusing on traditional measurements of uncertainty quantification like coverage, and emphasizing that the main contribution over prior work is an improved prior distribution for the Bayesian linear regression that serves as the explanation of each concept’s importance to the model behavior. That said, the empirical results in this paper are promising and I expect it will be publishable with some revisions.
> > >
> > > I will raise my score from 3 to 5 and my confidence from 3 to 4.

---

> > > > ### Author Response · Authors · 2023-11-22
> > > >
> > > > We thank the reviewer for acknowledging our response. Their feedback helped greatly improve the manuscript with new ablations and clarifications. We wish to clarify their follow-up questions.
> > > >
> > > > Regarding the comment on theoretical lapses.
> > > > > Thanks, I thought about this more and it almost makes sense, but I still feel that there is a flaw... Consider that, the proposed objective of w^Ts=0, could be satisfied for class i by placing positive weight w1 on s_1 and negative weight w2 on s_2, such that the inner product is 0.
> > > >
> > > > Thanks for the keen observation. We are omitting such solutions, i.e. $\vec{w}\epsilon\approx 0$ is satisified through large but negative values proportional to the noise, since we approximate $\vec{w}^T\epsilon\epsilon^T\vec{w}$ with $\vec{w}^T diag(\epsilon\epsilon^T)\vec{w}$ (please see towards the end of page 3). For the sake of clarity, imagine we have two dimensional $\vec{w}, \epsilon$, then we are approximating $w_1^2\epsilon_1^2+w_2^2\epsilon_2^2+2w_1w_2\epsilon_1\epsilon_2$ with $w_1^2\epsilon_1^2+w_2^2\epsilon_2^2$ ($w_1, w_2$ and $\epsilon_1, \epsilon_2$ are the first and second components of $\vec{w}, \epsilon$ respectively). Undesired solutions of the kind pointed by the reviewer are weeded out because the product between dimensions ($w_iw_j, i\neq j$) is ignored by the approximation.
> > > >
> > > > The prior we imposed $w\sim \mathcal{N}(0, \text{diag}(\epsilon\epsilon^T)^{-1})$ satisfies our constraint more naturally: "the linear estimator should not rely on concepts with large noise/uncertainty." The sampled values from the prior are such that their value ($\vec{w}[k]$) deviates from 0 only when the corresponding noise $\epsilon[k]$ is small.
> > > >
> > > > To further convince the desired effect of our prior, consider this worked out example with a single observation and 2 concepts. We have $\epsilon = [s_1^2, s_2^2]^T$ and $C_X=[m_1, m_2]^T, Y=y$. Simply evaluating the solution obtained from the posterior as shown in the Equation 1 of the paper, we get the posterior mean $\mu$ = $\beta\Sigma C_X Y$ = $[\kappa \frac{m_1}{s_1^2}, \kappa \frac{m_2}{s_2^2}]^T$ where $\kappa=y(m_1^2/s_1^2 + m_2^2/s_2^2 + 1/\beta)^{-1}$. We observe that the values of $\vec{w}$ after having observed an example is inversely proportional to the noise in that dimension (or concept), which is as desired.
> > > >
> > > > Regarding coverage.
> > > > > Then, the traditional way to measure this is via coverage, not via the error-rate correlation analysis of the appendix G.1.
> > > >
> > > > We agree that our evaluation of uncertainties in Appendix G is somewhat unorthodox. Unfortunately, we do not have access to ground-truth concept activations in order to evaluate our statement: "Our objective is to estimate m(x), s(x) such that the true concept activation value is in the range:m(x) ±s(x)." using a coverage-based measure.
> > > >
> > > > Nevertheless, we tried our best to evaluate the confidence regions. We expected that s(x) to account for both ambiguity (data uncertainty) and knowledge about the concept in the representation layer (epistemic uncertainty). Our experiments in Appendix G.1 confirmed that s(x) accounted for both the forms of uncertainty. Although, our evaluation may seem unorthodox, it is our best effort given the evaluation requirement.
> > > >
> > > > Regarding presentation related concerns.
> > > > We have rewritten Sections 3, 4 carefully to clarify the sources of uncertainty, expanded all derivations, explained the role of CLIP or  Oikarinen et al. (in estimating $cos(\alpha)$), added text with pointers to Appendices. Although the initial draft fell short on presentation, we believe the new draft improved the presentation significantly.
> > > >
> > > > We thank the reviewer once again for their time. We are very glad that our previous response resolved many of their original concerns and we truly appreciate the reviewer taking time to respond.

---

> > > > > ### Comment · Reviewer_xCUE · 2023-11-22
> > > > >
> > > > > Thanks for the additional response. I did not notice this consequence of the $\epsilon \epsilon^T \approx diag(\epsilon \epsilon^T)$ approximation before -- since this is quite important, if might be emphasized more heavily!
> > > > >
> > > > > I also did not immediately realize that issue with a coverage metric, and I agree that the chosen evaluation is reasonable. The lack of a ground truth in explainability research makes these kinds of evaluations very difficult. Perhaps this is a place for larger scale synthetic experiments, so that we can confirm the method's effectiveness against a ground truth.

---

> > > > > > ### Author Response · Authors · 2023-11-22
> > > > > >
> > > > > > We thank the reviewer once again for engaging in a discussion.
> > > > > >
> > > > > > We are glad our response clarified the concern regarding uncertainty evaluation. We will ensure to emphasize the role of approximation more carefully in the revised manuscript.

---

### Official Review · Reviewer_QofX · 2023-11-01

**Soundness:** 3 good
**Presentation:** 3 good
**Contribution:** 3 good
**Rating:** 6
**Confidence:** 4

**Summary:**

This paper observes that the lack of uncertainty estimation as part of models that integrated concepts is problematic. It then proposes learn a confidence interval for the concept scores that are learnt in traditional post-hoc concept bottleneck models. Overall, the scheme estimates concept activations along with a confidence interval for that particular activation. You then learn a linear predictor on top of the concept activation, however, the formulation here constrains the linear predictor to satisfy some particular properties. They then compare this uncertainty aware estimator to other post-hoc CBMs on a variety of synthetic and real-world datasets to demonstrate its favorable properties.

**Strengths:**

Overall, this work points out some challenges with current CBMs. Here is an overview of the key strengths of this paper.

- **Incomplete Concept Set, Concept Difficulty, and Shift**: This paper identifies two challenges, that are often true in practice that undermine the effective of standard concept activations. The paper then proposes a method to address this challenges. The core insight is that error interval allow us to know whether to trust the concept activations of the model. If the interval is large then caution needs to be taken.

- **Assessment in a Setting with Ground-Truth**: A big challenge with post-hoc interpretations is that it can be difficult to know when the method used to perform the post-hoc interpretation is reliable. In their experiments, the authors design a setting where the ground-truth is known, and  use it to assess their formulation along with several other baselines.

**Weaknesses:**

Overall, I think this paper has a nice formulation, but I was confused about certain aspects of the work. I detail them below.

- **Why focus on the post-hoc CBM setting?**: I understand the justification that it is difficult to get annotations for all of the training set. This is true and a known limitation, but I think the authors missed a chance to at least demonstrate the importance of the uncertainty estimation part of their work. Right now, I think the authors choose the model difficult setting, the post-hoc CBM. One way to show the effectiveness of this scheme is to first demonstrate in the setting where you have all training set annotations. Although I understand that when you use a small concept set to convert a black-box to CBM model, the challenges of dataset shift and incomplete concept set are exacerbated. For me, it would have been easier to digest the method independent of post-hoc CBMs. This is more of a suggestion.

- **Implications in Section 3**: Here the authors make a series of implications to arrive at distribution of weight vectors. I don't see how that holds clearly, especially the second implication. Why does a high probability bound on the dot product of the weight and noise imply that? What is the distributional assumption on the noise? Also, can the authors explain how they arrive at eqn 1? It looks like a gaussian identity, but I want to clarify.

- **Sparsifying the weights**: Instead of picking the threshold by hand, why can't the authors impose a lasso penalty here as is done for the simple baseline?

- **Section 3.1 is confusing**: Until here, there was nothing about multimodal functions/CLIP in the paper. But all of a sudden, it is incorporated to compute the mean and error function. I think you could do away with the CLIP discussion in that section and just call CLIP/Multimodal function an embedding function. Based on that description, I still don't understand how the mean and errors are computed. Proposition 1 discusses this, but why is this? It seems like this section adopts the notation of  Oikarinen et al. (2023). I would've thought that the procedure here can give an error estimate for any generic function that of the input to the concept activation? Is there a way to abstract away the specific form of the concept activation of Oikarinen et al. (2023) in this formulation? Stated differently, what is $\vec{m}(x)$, and $\vec{s}(x)$ for a standard CBM?

- **The problem with CLIP**: The formulation here ties in with CLIP intimately. It is clear that if you want to use something like this for say proteins, sequences, or other modalities, then CLIP will not help you. You'd want the equivalent of CLIP for the domain you are working with.

**Questions:**

I combined weaknesses and questions, so see the weaknesses section.

---

> ### Author Response · Authors · 2023-11-19
>
> We are grateful for the reviewer’s thoughtful feedback. We are very glad that the reviewer found our problem practical, motivation convincing, and evaluation setup strong.
>
> > Implications in Section 3: … Why does a high probability bound on the dot product of the weight and noise imply that? What is the distributional assumption on the noise? Also, can the authors explain how they arrive at eqn 1?
>
> Thanks for pointing out these issues. We added more detail in the main paper, added detailed derivations in the appendix to improve the readability of Section 3. We are happy to resolve any unresolved concerns.
>
> We did not make distributional assumption on the noise and fully derived the Eqn (1) in Appendix A.1.
>
> > Sparsifying the weights: Instead of picking the threshold by hand, why can't the authors impose a lasso penalty here as is done for the simple baseline?
>
> Depending on the choice of prior, we may not get a clean analytical form for the posterior on the weights as shown in Equation (1). Although our post-hoc sparsification of weights is not fully Bayesian and introduced an additional hyperparam, it worked satisfactorily for all our experiments.
>
> > Section 3.1 is confusing: … just call CLIP/Multimodal function an embedding function.
>
> We agree. We have revised the draft to emphasize the generality of our method to any multi-modal model.
>
> >…I … don't understand how the mean and errors are computed….
>
> We greatly appreciate your concern. In Section 3.1 of the revised draft, we have added intuition and filled in the missing details. Could you please take a look and let us know?
>
> > Is there a way to abstract away the specific form of the concept activation of Oikarinen et al. (2023) in this formulation?
>
> Thanks for the great observation, we agree. The role of Oikarinen et al. (2023) is only to inform us about how much information about the kth concept is encoded in the representation layer of the model-to-be-explained, which we denote by $cos(\alpha_k)$. We can use any method for estimation of $v_k$, we just used Oikarinen et al. (2023). In the revised draft, we changed the writing to abstract out specific form of estimation: Oikarinen et al. (2023)
>
> > The problem with CLIP: The formulation here ties in with CLIP intimately. …
>
> As the reviewer pointed out in their previous concern. U-ACE has very little to do with how and where the image and text embeddings: $g, g_{text}$ come from. Although our evaluation setup used CLIP, any other pretrained multi-modal model can be plugged-in very easily.  We understand that CLIP is not suitable in biomedical applications, but we can simply replace CLIP with an appropriately pretrained multi-modal model.
>
> > Why focus on the post-hoc CBM setting?
>
> We request the reviewer to please rephrase or elaborate? We did not understand the concern. We have concept annotation on full training data for all our settings. However, only TCAV and Simple have access to them. Y-CBM, O-CBM and U-ACE did not use the available concept annotations in any way.
>
> Regarding "chance to at least demonstrate the importance of the uncertainty estimation part of their work.",
> Appendix G.1 of the revised draft demonstrates the quality of uncertainty estimated by U-ACE.
>
>
> We thank the reviewer once again for their thorough assessment and are more than happy to engage further.

---

> > ### Comment · Reviewer_QofX · 2023-11-22
> > **Clarified my concerns**
> >
> > Thanks for the response. I think several of points that I was confused about are now mostly clear. I still need to go through the revision more carefully. However, on a first pass, my main concerns have been addressed. And a final revision can incorporate others. Overall, I think contribution here is valuable even if it is as a first steps towards incorporating uncertainty in concept bottleneck models. I can see future work proposing alternative schemes and/or improving the one here.
> >
> > One point that I didn't appreciate before until it was raised in another review is that it is actually challenging to evaluate the uncertainty estimates themselves when a clean function ground-truth behavior is not known. Overall, I think the contribution here is useful, and meaningful one to the literature on concept bottlenecks.

---

> > > ### Author Response · Authors · 2023-11-22
> > >
> > > We thanks the reviewer for acknowledging our response. We are very glad that their concerns are mostly resolved.

---

### Official Review · Reviewer_9tBC · 2023-11-02

**Soundness:** 3 good
**Presentation:** 2 fair
**Contribution:** 3 good
**Rating:** 6
**Confidence:** 3

**Summary:**

The authors propose a technique (U-ACE) for estimating concept relevance that is robust to misspecification of the concept set.  At a high level, their algorithm 1) estimates noise in concept activations, 2) takes it into account when estimating the concept-label weight matrix, 3) applies a sparsification step.  The paper also looks at the difference between this technique and a simpler linear fit in the context of linear models.  U-ACE is compared to other concept-based explainers (and concept-based models) on a variety of tasks.

**Strengths:**

**Originality**: The idea of incorporating concept uncertainty in CBEs is, to the best of my knowledge, novel.  It is however well aligned with recent work in CBMs (as mentioned in the related work).  The specific technique used here seems also to be novel.

**Quality**: The approach itself is well motivated.  Admittedly, I did not check the mathematical derivations.  The general idea is however sensible.  The research questions are well aligned with the key message.  The choice of competitors is also good.  The results seem to indicate that U-ACE is more robust to issues like vocabulary misspecification and data shift, which is good.

**Clarity**:  English is mostly good and the text is generally readable.

**Significance**: CBEs and CBMs are a pretty hot topic, and this paper touches on a number of very relevant aspects.  Uncertainty is definitely one element that is (at least in part) dismissed in the current literature, yet it is important for trustworthiness/faithfulness of explanations.

**Weaknesses:**

**Clarity**:  The text is definitely too dense in parts (especially Section 3).  Some paragraphs feel rushed and would enjoy a rewrite.  There are also some typos (I mentioned a few below).

**Significance**: [Q1] My understanding is that the implementation of m(x) and s(x) is model specific -- that is, the equation explain how to derive these quantities for image-text multimodal systems.  So, while the general setup is model-agnostic, the specific algorithm is likely not.

Minor issues
----

- p 2: There is a typo in the equation of v_k in page 2 (it should read \mathcal{D}^{k}_c).

- p 2: "a class of algorithms propose to train"

- p 3: "recall that K is number of concepts and L the number of labels" -> $K$, $L$.  Also in p 5.

- p 3: It wouldn't hurt to clarify the steps in the mathematical derivation.  Also, why is s(x) being averaged over?  Why can't it be used as-is?  If $\beta$ is the inverse variance of noise in observations, why is it being optimized?  Section 3 should be unpacked to facilitate understanding.  Right now it is unnecessarily opaque.

- p 6: "This baseline is used in the past"

- p 6, plots: "Fration"

- p 8: "since model-to-be-explained"

- p 8: "We note that U-ACE generated explanations are more convincing over O-CBM or Y-CBM."

**Questions:**

Q1.  As I mentioned in the weakness section, I am under the impression the implementation is model specific.  Could you please confirm this?  If so, how should m(x) and s(x) be estimated for other models?

I am willing to increase my score provided the authors clarify this point.

---

> ### Author Response · Authors · 2023-11-19
>
> We are grateful for the reviewer’s time and efforts. We are very glad that the reviewer found uncertainty modelling  as novel and important, problem well-motivated and topical, evaluation setup well aligned with the key message, and our results convincing.
> > Clarity: The text is definitely too dense in parts (especially Section 3). Some paragraphs feel rushed and would enjoy a rewrite. There are also some typos (I mentioned a few below).
> > Minor issues…
>
> Thanks for pointing out these issues, all the minor issues are now resolved in the revised draft. We have also rewritten parts of Section 2, 3 to improve clarity and readability.
>
> > It wouldn't hurt to clarify the steps in the mathematical derivation. Also, why is s(x) being averaged over? Why can't it be used as-is?
>
> We did it for the ease of analysis. We get a clean closed form if we average over s(x).
>
> > If is the inverse variance of noise in observations, why is it being optimized?
>
> Thanks for raising this issue. We added the following passage to the paper. We could directly set the inverse of $\beta$ approximately 0 since there is no noise on the observations: Y. Instead of setting $\beta$ to an arbitrary large value, we observed better explanations when we allowed the tuning algorithm to find a value of $\beta, \lambda$ to balance the evidence and noise.
>
> > Section 3 should be unpacked to facilitate understanding.
>
> Thank you very much for pointing out the problems. We have rewritten Section 3 to improve its readability.
>
> > … I am under the impression the implementation is model specific. Could you please confirm this? If so, how should m(x) and s(x) be estimated for other models?
>
> Our implementation is not specific to any model. We only assume access to the representation layer and the logits of the model-to-be-explained. We did not make any specific assumptions about the multi-modal model. CLIP can be easily replaced with any other pretrained multi-modal model in a plug-and-play manner.
>
> A pretrained multi-modal model defines g, $g_{text}$ of Section 3.1 and any new model-to-be-explained must define ‘f’ (a mapping from the input to the representation and logit). The rest of the details must follow for computation of m(x), s(x) as described in Proposition 1.
>
> We have revised the draft to emphasize the generality of our proposal to any (neural-network-based) model and any multi-modal model.
>
> Please let us know if we have misinterpreted your question or you found our response not convincing.
>
> We thank the reviewer once again for encouraging feedback and careful assessment.

---

> > ### Comment · Reviewer_9tBC · 2023-11-22
> > **Reply to authors**
> >
> > Thank you for addressing my questions.  I also appreciate the changes made to the manuscript.  I will give them some thought before deciding whether to change my score.  In the meantime, I have a couple of remarks:
> >
> > - I guess the argmax in the second paragraph of Section 2 should be an argmin instead.
> >
> > - I feel the "noun-colon-symbol" structure frequently used in the text is unnecessary, non-standard, and slightly slows down reading.  I suggest to get rid of the colons altogether.  This would improve readability.

---

> ### Author Response · Authors · 2023-11-22
>
> We thank the reviewer for their response.
>
> Both the remarks are addressed in the (just) revised manuscript. Thanks for the great catch! We agree that we used "noun-colon-symbol" overly in the previous draft.
>
> We are more than happy to answer any unresolved concerns. We thank the reviewer once again for engaging in discussion with us.

---

### Author Response · Authors · 2023-11-19

We are elated to receive passionate and detailed feedback from the reviewers. We are very grateful for their time and efforts. We are glad that the reviewers found the motivation clear and appreciated the need to model uncertainty as novel, interesting and/or sensible. We addressed specific questions from each reviewer in our response and addressed the common issue with presentation clarity by carefully editing the Sections 3, 4.

We made the following changes to the draft.
* We fixed all the typos and missing details (especially in Section 3, 4).
* We fixed the text to make clear the following points in Section 3.1: (a) CLIP can be replaced with any other pretrained multi-modal model, (b) source of uncertainty that U-ACE estimates, (c) relegated the role of the specific method for estimating concept activation vectors.
* We added multiple ablation experiments to Appendix G further motivating the design of U-ACE. In Appendix G.1, we evaluated the uncertainty on concept activations estimated by U-ACE and compared with other simpler methods or existing baselines for estimation of uncertainty. In Appendix G.2, we evaluated the need for U-ACE’s uncertainty-aware prior by comparing U-ACE with simple Bayesian regression. In Appendix G.3, we presented sensitivity of the baselines: Y-CBM, O-CBM to regularization strength hyperparameter.

The newly added text to the draft is marked in blue.

We are happy to answer any other unresolved concerns. We thank the reviewers once again for their thoughtful assessment of our work.

---

> ### Author Response · Authors · 2023-11-23
>
> Gentle reminder that the author-discussion deadline is in few hours. We are happy to answer any unresolved concerns. Please consider championing our paper if you like it.
>
> We thank the reviewers once again for immensely useful feedback/discussion so far.

---

### Meta-Review · Area_Chair_y7Tp · 2023-12-10

**Metareview:**

This paper proposes a new technique called U-ACE to generate concept annotations using a pre-trained multimodal model (e.g., CLIP). The authors describe how U-ACE can be used to generate concept-based explanations like TCAV in a way that does not require concept annotations and can account for the inherent uncertainty in explanations. The paper includes a variety of experiments to highlight the value of this technique -- showing that it is robust to misspecification and data set shift.

*Strengths*

- Topic: The proposed method could provide an off-the-shelf solution to automatically generating concept annotations. This is a key barrier to the adoption for a large number of techniques in interpretable and explainable deep learning -- ranging from TCAV to concept bottleneck models.

*Weaknesses / What's Missing*

- Significance: The primary technical contribution of this work -- as noted by all reviewers -- is a novel technique that can be used to generate concept annotations. The manuscript hints at two ways that this technique could be used in practice: (1) to evaluate the uncertainty of concept explanations, and (2) to build "label-free" concept bottleneck models. The paper primarily focuses on the first use case, choosing to focus on the second one. This is unfortunate -- as it is far more difficult to make a compelling case that concept explanations should be unaware in this case. The key issue in this case is that we need to be able to see how the reliable uncertainty estimates improve performance in a concrete task e.g., to debug the model or to flag predictions that require human review. In contrast, it would be far easier to make a compelling case for uncertainty in explanations if we could see its impact on e.g., the performance of an off-the-shelf CBM.

- Scope/Clarity/Limitations: The proposed technique makes clever use of a pre-trained multimodal model (e.g.,  CLIP). This is a sensible limitation -- albeit one that merits further discussion. In this case, the weakness is a lack of clarity to key questions that would affect the significance of the work. What are the kinds of applications where we could make use of this technique? And do they correspond to those where we could use CBMs in the first place? To reap the benefits of this technique in concept bottlenecks, for example, we would ideally need the technique to annotate concepts that are deterministic and human-verifiable -- i.e., where the noise pertains to unreliable annotation rather than the lack of uncertainty.

**Justification For Why Not Higher Score:**

The scores are lukewarm and the paper has no champion for acceptance. There is significant room for improvement by addressing the changes above.

**Justification For Why Not Lower Score:**

N/A

---

### Decision · Program_Chairs · 2024-01-16

Reject